# A Paradigm Shift in Machine Translation: Boosting Translation Performance of Large Language Models

**Haoran Xu♠, Young Jin Kim♡, Amr Sharaf♡, Hany Hassan Awadalla♡**

♠Johns Hopkins University, ♡Microsoft

hxu64@jhu.edu
{youki,amrsharaf,hanyh}@microsoft.com

## Abstract

Generative Large Language Models (LLMs) have achieved remarkable advancements in various NLP tasks. However, these advances have not been reflected in the translation task, especially those with moderate model sizes (i.e., 7B or 13B parameters), which still lag behind conventional supervised encoder-decoder translation models. Previous studies have attempted to improve the translation capabilities of these LLMs, but their gains have been limited. In this study, we propose a novel fine-tuning approach for LLMs that is specifically designed for the translation task, eliminating the need for the abundant parallel data that traditional translation models usually depend on. Our approach consists of two fine-tuning stages: initial fine-tuning on monolingual data followed by subsequent fine-tuning on a small set of high-quality parallel data. We introduce the LLM developed through this strategy as Advanced Language Model-based trAnslator (**ALMA**). Based on LLaMA-2 (Touvron et al., 2023b) as our underlying model, our results show that the model can achieve an average improvement of more than 12 BLEU and 12 COMET over its zero-shot performance across 10 translation directions from the WMT'21 (2 directions) and WMT'22 (8 directions) test datasets. The performance is significantly better than all prior work and even superior to the NLLB-54B model (NLLB TEAM et al., 2022) and GPT-3.5-text-davinci-003, with only 7B or 13B parameters. This method establishes the foundation for a novel training paradigm in machine translation. [1]

## 1 Introduction

Generative (decoder-only) large language models (LLMs) such as GPT models (Brown et al., 2020; OpenAI, 2023), PaLM (Chowdhery et al., 2022), OPT (Zhang et al., 2022), BLOOM (Scao et al., 2022), LLaMA (Touvron et al., 2023a;b), and others have exhibited remarkable capabilities across various NLP tasks. However, for the translation task, only very large models such as GPT-3.5 and GPT-4 can rival the supervised encoder-decoder state-of-the-art (SoTA) models like NLLB (NLLB TEAM et al., 2022), while they still fall short in translation for low-resource languages (Hendy et al., 2023; Jiao et al., 2023). The discrepancy becomes more evident when comparing other LLMs with traditional translation models (Zhu et al., 2023a). For instance, the OPT-175B model trails behind the NLLB-1.3B model by an average of more than 15 BLEU (Papineni et al., 2002) points for languages within the Indo-European-Romance family. The gap is even larger in smaller LLMs; for example, XGLM (Lin et al., 2021), with a parameter size of 7B, lags behind the NLLB-1.3B by a substantial 30 BLEU points (Zhu et al., 2023a). Therefore, there is an urgent need to narrow this performance gap between LLMs and conventional SoTA models.

As exemplified by NLLB-1.3B, traditional machine translation models demonstrate proficiency in producing high-quality translations with a small number of parameters. By extension, smaller LLMs

---

[1]We release our code and models at: https://github.com/fe1ixxu/ALMA.

should similarly possess the capability to adeptly manage the translation task. Recent research has sought to enhance translation performance by commencing with smaller LLMs (Yang et al., 2023; Zeng et al., 2023; Chen et al., 2023; Zhu et al., 2023b; Li et al., 2023; Zhang et al., 2023b), especially 7B or 13B parameters. Nevertheless, the achieved improvements remain modest and limited. As depicted in Figure 1, contemporary studies such as Balyling (Zhang et al., 2023b) and BigTranslate (Yang et al., 2023), which use LLaMA as their backbone, exhibit a maximum increment of 3 to 4 BLEU or COMET in relation to the zero-shot performance of LLaMA on the WMT'22 test set (8 directions).[2] While these gains represent promising research direction for smaller LLMs in the translation task, a significant performance chasm persists when benchmarked against very large LLMs such as GPT-3.5-`text-davinci-003` and SoTA translation models such as NLLB-54B. We posit that the modest translation gains observed in prior studies can be ascribed to an unsuitable training recipe.

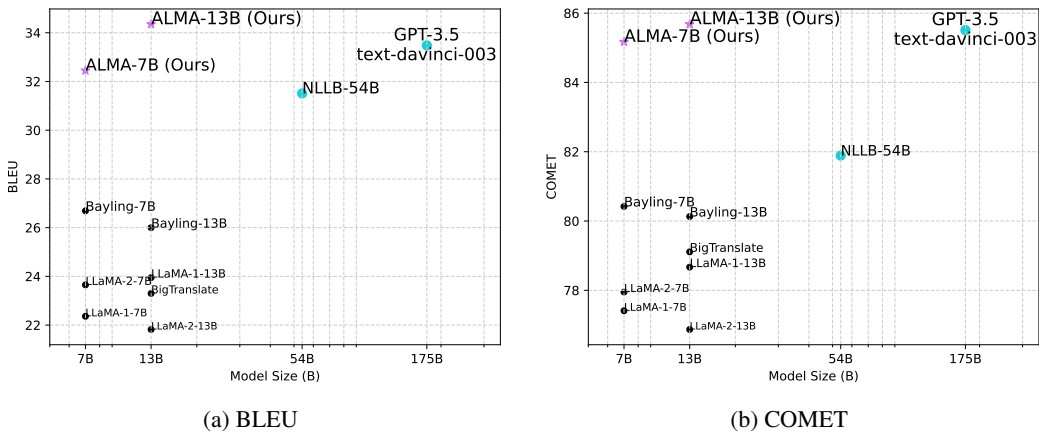

(a) BLEU
(b) COMET

Figure 1: Translation performance of contemporary decoder-only LLM translation systems based on LLaMA (Yang et al., 2023; Zhang et al., 2023b), and zero-shot performance of LLaMA, for the WMT'22 test data across 8 directions (translating to or from English for German, Czech, Chinese, and Russian). Benchmark comparisons also include two leading translation models, NLLB-54B and GPT-3.5-`text-davinci-003`. Our systems, developed on LLaMA-2 with 7B and 13B parameters, surpass previous models by an impressive margin of nearly 10 BLEU and 7 COMET. Furthermore, they even slightly outperform GPT-3.5 and NLLB-54B on average.

We hypothesize that an efficacious training recipe ought to follow two stages: *learning general multilingual linguistic knowledge* and *inducing (instructing) models toward translation generation*. Consequently, we propose a two-stage fine-tuning approach and introduce the LLM developed through this strategy as **A**dvanced **L**anguage **M**odel-based tr**A**nslator (**ALMA**). Specifically, given most LLMs are trained on English-dominant data, the first stage is fine-tuning non-English monolingual data to enhance the model's proficiency in other languages involved in the translation task. Secondly, drawing inspiration from the recognized significance of data quality in other applications (Zhou et al., 2023; Maillard et al., 2023; Gunasekar et al., 2023), we fine-tune the model with a small amount of high-quality parallel data.

Our main contributions are summarized as follows:

**Diminished Necessity of Parallel Data** Traditional translation frameworks rely on large amounts of parallel data, which may lead to a false impression that such data is essential for the translation task with LLMs. Prior studies have fine-tuned LLMs with datasets containing over 300M parallel instances (Yang et al., 2023). However, our empirical evaluations suggest that this strategy may not be optimal, and even harm the translation capabilities of LLMs.

**LLM Via A New Training Recipe: ALMA** We introduce a novel two-stage fine-tuning method for translation with decoder-only LLMs. Leveraging LLaMA-2 as the base model, we attain an average improvement of more than 12 BLEU and COMET scores over its zero-shot performance

---

[2]All COMET scores in the paper is COMET-22 (`Unbabel/wmt22-comet-da`) (Rei et al., 2022).

across 10 translation directions from WMT'21 and WMT'22 test datasets. Notably, the performance surpasses all previous work and is even better than the NLLB-54B model and GPT-3.5-text-davinci-003.

**Efficient Computational Cost** Our ablation study reveals both stages are crucial factors for achieving large improvements. The most computationally intensive part is monolingual data fine-tuning, however, we show that only fine-tuning 1B monolingual tokens is sufficient to have comparable performance to NLLB-54B in 10 translation directions, which only requires around 18 hours to complete with 16 MI200 GPUs.

## 2 PRELIMINARY

### 2.1 TASK DEFINITION

We consider a decoder-only transformer model parameterized by $\theta$ for machine translation. Let $\mathbf{x}$ represent the source sentence and $\mathbf{y}$ its corresponding target sentence. We utilize a fixed prompt template, denoted as $\mathcal{I}$, to guide the model in generating translation. The log-likelihood loss of the parallel sentence $(\mathbf{x}, \mathbf{y})$ with regard to the model parameters $\theta$ can be formulated as follows:

$$\mathcal{L}_{\text{NLL}}(\mathbf{x}, \mathbf{y}, \theta) = -\log P(\mathbf{y}|\mathbf{x}, \mathcal{I}; \theta) \tag{1}$$

$$= -\sum_{t=1}^{T} \log P(y_t|\mathbf{y}_{<t}, \mathbf{x}, \mathcal{I}; \theta), \tag{2}$$

where $T$ is length of the target sentence, and $y_t$ is the $t$-th target token. The loss is a standard causal language modeling (CLM) loss, which predicts the next token based on prior tokens. We use the same sentence-level translation prompt template suggested by Hendy et al. (2023), and illustrate the prompt and the model input/target in Figure 2. Note that we do not compute the loss for the prompt template and source sentence during training (Zhang et al., 2023a). In Appendix A, we show that CLM is more suitable for the translation task compared with other modeling methods, such as prefix language modeling (Wang et al., 2022) and mixture-of-denoisers (Tay et al., 2022a).

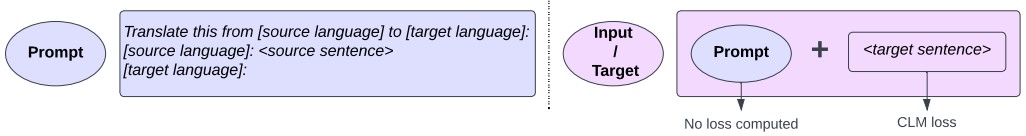

Figure 2: The prompt used for training and evaluation. *[source language]* and *[target language]* represent the full name of the language, e.g., Translate this from German to English. Note that we do not compute loss for the prompt.

### 2.2 A BACKBONE LLM FOR TRANSLATION

We seek a robust LLM to serve as our foundational model. With the recent emergence of numerous LLMs, we prioritize evaluating the zero-shot translation performance of these models before delving into optimal training recipes. As most of these models provide a 7B version, our comparative analysis centers on this magnitude: OPT-7B (Zhang et al., 2022), Falcon-7B (Almazrouei et al., 2023), BLOOM-7B (Scao et al., 2022), MPT-7B (MosaicML, 2023), LLaMA-1-7B (Touvron et al., 2023a), and LLaMA-2-7B (Touvron et al., 2023b). We additionally present results from GPT-3.5-text-**d**avinci-003 (hereinafter referred to as **GPT-3.5-D**) and GPT-3.5-**t**urbo-0301 (hereinafter referred to as **GPT-3.5-T**) to show the performance gap.[3]

**Zero-Shot Evaluation** We conduct zero-shot evaluations on 5 English-centric language pairs, considering both from English and to English directions: German (de), Czech (cs), Icelandic (is), Chinese (zh) and Russian (ru), where Icelandic test data is from WMT'21 and the others are from WMT'22. We choose these test dataset because they are the recent and less likely

---

[3]https://beta.openai.com/docs/model-index-for-researchers

to overlap the training data used by LLMs, and importantly, they have high-quality data to avoid problems of "translationese" (Zhang & Toral, 2019). The beam size is 5. We report sacre-BLEU (`zh` tokenizer for Chinese and `13a` for the others) (Post, 2018). We also report COMET (`Unbabel/wmt22-comet-da`) (Rei et al., 2022) because BLEU only reflects the degree of lexical match. In this paper, We rely more on COMET than BLEU due to its better alignment with human evaluations (Freitag et al., 2022).[4]

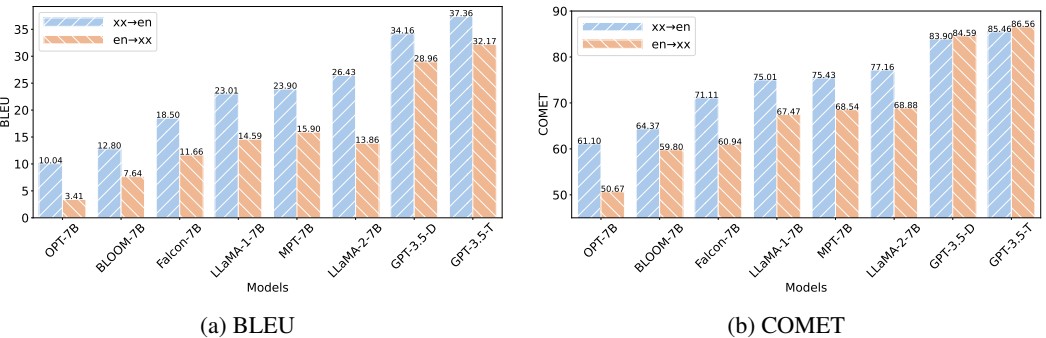

(a) BLEU        (b) COMET

Figure 3: Averaged zero-shot translation performance on 10 directions: cs↔en, de↔en, is↔en, zh↔en, ru↔en, where is↔en is from WMT'21 test data and the others from WMT'22 test data.

**LLM Translation Performance** The overall results for the LLMs are presented in Figure 3, with scores averaged across five languages for translations to and from English. Among the 7B LLMs, LLaMA-2-7B exhibits superior performance translating into English, while MPT-7B leads in translations out of English, as measured by BLEU. However, when evaluated with COMET, LLaMA-2-7B wins in both directions. We show the numeric results in Appendix B. Consequently, we select LLaMA-2-7B and MPT-7B for further investigation into the necessity of parallel data for LLMs.

## 3 Do LLMs Have an Appetite for Parallel Data?

Conventional machine translation training predominantly relies on utilizing large volumes of parallel datasets within encoder-decoder frameworks. This trend is not confined to training models from scratch but also pertains to strategies that fine-tune pre-trained LLMs, often involving millions of parallel sentences (Rothe et al., 2020; Liu et al., 2020; Xu et al., 2021; 2023; Yang et al., 2023). In this section, we examine whether the recently proposed decoder-only LLMs retain a dependence on substantial parallel data and adhere to the traditional training paradigm.

### 3.1 Experimental Design

Following Section 2.2, our focus narrows to fine-tuning LLaMA-2-7B and MPT-7B. To allow for a deep analysis, we concentrate on one language pair, English→Russian (en→ru). We opted for a language pair that is translating out of English and to a non-Latin language, since those categories show larger gaps with SoTA models in our initial investigation in Section 2.2. We use the clean data filtered from 75M parallel sentences from Hendy et al. (2023) and split the data size in 5 levels: 10K, 100K, 1M, 5M, and 20M. We use the same prompt template and training scheme as described in Section 2.1, and train the model by updating all parameters. Detailed training settings can be found in Appendix C.

### 3.2 Observations

The fine-tuning results for LLaMA-2-7B and MPT-7B at each data size step are presented in Figure 4. Additionally, we benchmark these against the performance of the NLLB-54B model to show the disparity with one of the SoTA multilingual translation models.

---

[4]According to Freitag et al. (2022), COMET holds the 2-nd position in alignment with human ratings, whereas BLEU is situated at the 19-th spot among 20 metrics.

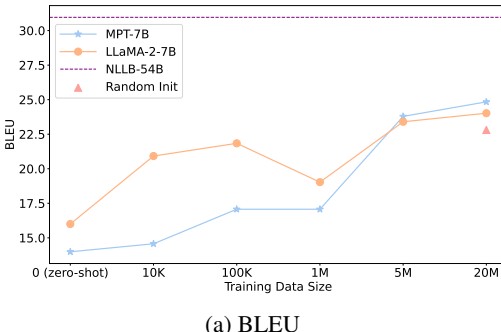 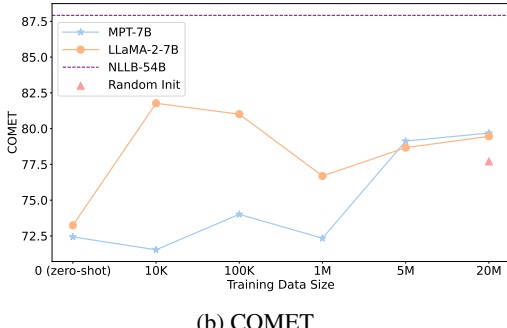

(a) BLEU  (b) COMET

Figure 4: BLEU and COMET scores obtained during the fine-tuning of MPT-7B and LLaMA-2-7B across each data step for en→ru. Additionally, we present the results for NLLB-54B and a 7B model trained from scratch. A notable decline in LLaMA-2-7B's COMET score suggests that substantial parallel data might dilute its pre-existing knowledge.

**Small Training Data Is Enough** According to COMET, there is a notable difference in the curve of LLaMA-2-7B and MPT-7B: LLaMA-2-7B peaks with 10K and 100K training data before experiencing a decline, while MPT-7B exhibits continuous improvement with more training data. LLaMA-2-7B requires only limited training examples (10K and 100K) to achieve competent translation. However, a surplus of examples (5M or 20M) seems to dilute its existing knowledge in Russian. Conversely, MPT-7B, potentially due to its inherently weaker translation capability, exhibits improved performance with an increase in training data. This may suggest that LLaMA-2 or other well-trained LLMs may not necessitate substantial parallel data.

**Large Parallel Data Wash Out the Knowledge** Both LLMs eventually achieve similar BLEU and COMET with 20M training data, regardless of their performance on smaller data. We hypothesize that this phenomenon is caused by catastrophic forgetting (French, 1999; Kirkpatrick et al., 2017), suggesting that too many parallel data wash out the pre-existing knowledge. To validate this hypothesis, we consider an extreme case: training the model from scratch using 20M data, thereby erasing all prior knowledge.[5] As expected, it tends up with a similar performance in both BLEU and COMET evaluations (triangle in Figure 4), strengthening our speculation regarding the dilution of LLM's intrinsic knowledge with extensive data training.

**Beyond BLEU** COMET reveals a decline in translation performance for LLaMA-2-7B as the amount of parallel data increases, a trend not captured by BLEU which shows an increase. This discrepancy arises since BLEU primarily evaluates lexical overlap, and the extensive WMT training data, being similar in domain to the test set, likely enhances this measure. This highlights the necessity of utilizing additional metrics (like COMET) for a comprehensive evaluation of translation.

From our observations, LLaMA-2 (potentially other well-trained LLMs) should not adopt the same training approach as earlier models——whether randomly initialized or pre-trained—that rely heavily on vast amounts of training data.

## 4 A New Training Recipe

We demonstrate that LLMs like LLaMA-2-7B do not voraciously consume parallel data. We introduce a novel training strategy that markedly enhances translation performance without relying heavily on parallel data. The recipe comprises two stages: continuous **monolingual data fine-tuning** and **high-quality parallel data fine-tuning**. After applying our training recipe to LLMs, we name the resulting model as **ALMA** (**A**dvanced **L**anguage **M**odel-based tr**A**nslator).

**Monolingual Data Fine-tuning** LLMs like LLaMA are pre-trained on English-dominated corpora. This potentially explains their inadequate translation performance which necessitates cross-lingual capabilities. To ameliorate this, our first stage is fine-tuning LLMs with monolingual data of non-English languages involved in translation tasks, enhancing their proficiency in these languages. Note

---

[5]We initialize parameters randomly based on the LLaMA-2-7B model, but use the same vocabulary.

that we also add English monolingual data during fine-tuning to prevent English knowledge forgetting. Previous studies also offer some clues that monolingual data help in translation. For instance, Tan et al. (2023) utilizes a monolingual target corpus to bridge the gap in translation mismatches caused by domain discrepancies. BigTranslate (Yang et al., 2023) and PolyLM (Wei et al., 2023) use a huge amount of Chinese monolingual data and improve translation to or from Chinese. Furthermore, Li et al. (2023) utilizes monolingual generation instructions to improve translation. In Section 6.1, we show that utilizing small monolingual data and modest computational cost (e.g., 1B monolingual tokens mixed by 6 languages and fine-tuning under 18 hours), can facilitate significant improvements in 10 translation directions. Note that we employ full-weight fine-tuning at this stage.

**High-Quality Data Fine-tuning** Drawing on insights from Section 3.2 that LLMs may require only small parallel data, coupled with previous research emphasizing training data quality (Zhou et al., 2023; Maillard et al., 2023; Gunasekar et al., 2023), we fine-tune the model using a small, yet high-quality parallel dataset in this stage. To ensure the data quality, we collect human-written datasets from WMT test data and Flores-200 (NLLB TEAM et al., 2022) development and test sets. Here, we explore both full-weight and lightweight Low-Rank Adaptation (LoRA) (Hu et al., 2022; Mangrulkar et al., 2022) fine-tuning, where we apply LoRA to the down-projection layer in each feed-forward network.

## 5 EXPERIMENTS

### 5.1 DATA

For our parallel training data, we collect human-written test datasets from WMT'17 to WMT'20, plus the development and test sets from Flores-200 (NLLB TEAM et al., 2022), resulting in a total of 58K training examples across all languages. For the test data, we still use the same 10 translation directions to be consistent with our study in Section 2: cs↔en, de↔en, is↔en, zh↔en, ru↔en, where is↔en is from WMT'21 and the others are from WMT'22. Test data in WMT'21 (except for is) is used for the development dataset (a total of 8K parallel sentences).[6] The monolingual dataset is sourced from OSCAR (Ortiz Su'arez et al., 2019; Kreutzer et al., 2022). We mix the monolingual data and fine-tune the model with a sampling ratio of 20%, 14%, 8%, 19%, 22%, and 17% respectively for de, cs, is, zh, ru and en. We explain the reasoning behind the sampling ratios and show the detailed parallel data information in Appendix D.

### 5.2 TRAINING SETUP

We train the model in a many-to-many multilingual translation manner, and use LLaMA-2-7B (or 13B) as our backbone model given its best zero-shot performance. Our two-stage fine-tuning process yields two model types, differentiated based on the utilization of LoRA:

**ALMA-7B/ALMA-13B** *Full-Weight* fine-tuning on monolingual data followed by *Full-Weight* fine-tuning on high-quality parallel data for LLaMA-2-7B or -13B models.

**ALMA-7B-LoRA/ALMA-13B-LoRA** *Full-Weight* fine-tuning on monolingual data followed by *LoRA* fine-tuning on high-quality parallel data for LLaMA-2-7B or -13B models.

If using LoRA, the LoRA rank is 16 and only updates 0.1% parameters (7.7M for 7B and 12M for 13B model). Both monolingual data fine-tuning and human-written data fine-tuning basically share the same hyperparameter settings. Specifically, we fine-tune LLaMA-2 with a batch size of 256, a warm-up ratio of 0.01, and a sequence containing a maximum of 512 tokens. For monolingual data fine-tuning, we train the LLaMA-2-7B up to 20B tokens and LLaMA-2-13B up to 12B tokens. However, it is very likely that the model would be better in translation with more monolingual data fine-tuning. For human-written data fine-tuning, we train the model for 2 epochs (enough to see a clear convergence) and pick the best model with the lowest validation loss. For both stages, we adopt deepspeed (Rasley et al., 2020) to accelerate our training.

---

[6]There is no development dataset for Icelandic.

## 5.3 BASELINES

We evaluate our method against two baseline categories. First, we consider prior studies with the goal aligning with ours: leveraging LLMs for translation. Secondly, we benchmark against the current SoTA translation models. It's worth noting that this comparison isn't entirely fair due to discrepancies in training data and model architectures (e.g., 175B GPT-3.5 vs. our 7B models). Nevertheless, utilizing the same test set provides insights into our model's current standing.

**Prior Similar Work** We compare our model with BigTranslate (Yang et al., 2023), which extends LLaMA-1-13B to over 100 translation directions; TIM (Zeng et al., 2023), which uses correct and incorrect examples to help LLM to learn translation; SWIE (Chen et al., 2023), which improves LLM in translation via instruction augmentation; and BayLing (Zhang et al., 2023b), which uses interactive translation instructions. Given that the same test data and evaluation metrics are utilized, we directly report BLEU and COMET from their papers (except for BigTranslate, we assess their released model using the prompt they provided).

**SoTA Models** We consider the NLLB-54B model, which is the largest and best translation model released in the NLLB family (NLLB TEAM et al., 2022); and the zero-shot performance of GPT-3.5-text-davinci-003 (**GPT-3.5-D**) and GPT-3.5-turbo-0301 (**GPT-3.5-T**). Additionally, we present the zero-shot results for GPT-4.[7]

| Models | de BLEU | de COMET | cs BLEU | cs COMET | is BLEU | is COMET | zh BLEU | zh COMET | ru BLEU | ru COMET | Avg. BLEU | Avg. COMET |
|---|---|---|---|---|---|---|---|---|---|---|---|---|
| *SoTA Models* | | | | | | | | | | | | |
| NLLB-54B | 34.50 | 86.45 | 37.60 | 90.15 | 24.15 | 81.76 | 27.38 | 78.91 | 30.96 | 87.92 | 30.92 | 85.04 |
| GPT-3.5-D, zero-shot | 31.80 | 85.61 | 31.30 | 88.57 | 15.90 | 76.28 | 38.30 | 85.76 | 27.50 | 86.74 | 28.96 | 84.59 |
| GPT-3.5-T, zero-shot | 34.40 | 87.00 | 32.92 | 90.17 | 18.74 | 81.04 | 44.90 | 87.00 | 29.90 | 87.60 | 32.17 | 86.56 |
| GPT-4, zero-shot | 35.38 | 87.44 | 34.53 | 90.77 | - | - | 43.98 | 87.49 | 30.45 | 88.87 | - | - |
| *Prior Similar Studies* | | | | | | | | | | | | |
| TIM-BLOOMZ-7B | 20.63 | 74.16 | - | - | - | - | 37.20 | 84.89 | - | - | - | - |
| TIM-LLaMA-1-7B | 25.59 | 82.56 | - | - | - | - | 19.33 | 75.46 | - | - | - | - |
| SWIE-BLOOMZ-7B | 21.83 | 75.17 | - | - | - | - | 36.88 | 84.53 | - | - | - | - |
| SWIE-LLaMA-1-7B | 27.21 | 82.36 | - | - | - | - | 31.24 | 80.63 | - | - | - | - |
| BigTranslate-13B | 21.48 | 78.81 | 20.67 | 80.65 | 2.28 | 35.56 | 28.56 | 81.31 | 17.66 | 78.21 | 18.13 | 70.91 |
| Bayling-13B | 25.62 | 82.69 | 16.43 | 78.22 | - | - | 37.92 | 84.62 | 12.77 | 71.01 | - | - |
| *Our Recipe with Backbone Model: LLaMA-2-7B* | | | | | | | | | | | | |
| LLaMA-2-7B, zero-shot | 19.00 | 76.39 | 16.02 | 79.13 | 1.33 | 43.83 | 16.97 | 71.80 | 16.00 | 73.24 | 13.86 | 68.88 |
| ALMA-7B (Ours) | 30.31 | 85.59 | 29.88 | 89.10 | 25.71 | 85.52 | 36.48 | 85.05 | 27.09 | 87.17 | 29.89 | 86.49 |
| ALMA-7B-LoRA (Ours) | 30.16 | 85.45 | 30.17 | 89.05 | 25.19 | 85.44 | 36.47 | 84.87 | 26.93 | 87.05 | 29.78 | 86.37 |
| *Our Recipe with Backbone Model: LLaMA-2-13B* | | | | | | | | | | | | |
| LLaMA-2-13B, zero-shot | 13.69 | 75.55 | 0.87 | 68.57 | 2.36 | 38.47 | 30.00 | 79.70 | 0.59 | 63.84 | 9.50 | 65.23 |
| ALMA-13B (Ours) | 31.37 | 85.45 | 31.12 | 89.42 | 26.67 | 85.85 | 39.05 | 85.76 | 28.76 | 87.50 | 31.39 | 86.80 |
| ALMA-13B-LoRA (Ours) | **31.47** | **85.62** | **32.38** | **89.79** | **26.68** | **86.08** | **39.84** | **85.96** | **28.96** | **87.53** | **31.87** | **87.00** |

Table 1: The overall results in en→xx. ALMA models significantly outperform all prior similar studies and are comparable to SoTA models. We categorize BLEU and COMET scores into three groups: scores that are more than 10 points below the higher value of GPT-4/GPT-3.5-T are emphasized in dark red boxes, those that are more than 5 points below are emphasized in shallow red boxes, and all other scores are emphasized in green boxes. **Bold numbers** represent the highest scores among ALMA models and prior similar studies.

## 5.4 RESULTS

We show our main results of en→xx and xx→en respectively in Table 1 and 2. In summary, our best system (ALMA-13B-LoRA) outperforms all previous studies, NLLB-54B, and GPT-3.5-D, while it marginally underperforms compared to GPT-3.5-T and GPT-4.

**Comparing With LLaMA-2 Zero-Shot** For all 10 translation directions and both 7B and 13B models, LLaMA-2 trained by our recipe significantly outperforms its original zero-shot performance. For instance, ALMA-7B achieves +16.12 BLEU and +17.61 COMET for en→xx on average. It is worth noting that LLaMA-2-13B suffers from the off-target issue in en→xx zero-shot translation. However, it can be substantially alleviated by few-shot in-context learning (Brown et al., 2020), but still largely lag behind our methods (e.g., over 10 BLEU and COMET when translating from English). We discuss this further in Appendix E.

---

[7]GPT-4 results are sourced from Zhang et al. (2023b).

| Models | de | | cs | | is | | zh | | ru | | Avg. | |
|---|---|---|---|---|---|---|---|---|---|---|---|---|
| | BLEU | COMET | BLEU | COMET | BLEU | COMET | BLEU | COMET | BLEU | COMET | BLEU | COMET |
| *SoTA Models* | | | | | | | | | | | | |
| NLLB-54B | 26.89 | 78.94 | 39.11 | 80.13 | 23.09 | 71.66 | 16.56 | 70.70 | 39.11 | 81.88 | 28.95 | 76.66 |
| GPT-3.5-D, zero-shot | 30.90 | 84.79 | 44.50 | 86.16 | 31.90 | 82.13 | 25.00 | 81.62 | 38.50 | 84.80 | 34.16 | 83.90 |
| GPT-3.5-T, zero-shot | 33.10 | 85.50 | 47.20 | 87.30 | 37.50 | 85.50 | 26.60 | 82.90 | 42.40 | 86.10 | 37.36 | 85.46 |
| GPT-4, zero-shot | 33.87 | 85.62 | 48.67 | 87.43 | - | - | 27.20 | 82.79 | 43.51 | 86.18 | - | - |
| *Prior Similar Studies* | | | | | | | | | | | | |
| TIM-BLOOMZ-7B | 24.31 | 77.65 | - | - | - | - | 23.42 | 79.50 | - | - | - | - |
| TIM-LLaMA-1-7B | 27.91 | 82.80 | - | - | - | - | 19.33 | 75.46 | - | - | - | - |
| SWIE-BLOOMZ-7B | 25.95 | 78.80 | - | - | - | - | 23.40 | 79.36 | - | - | - | - |
| SWIE-LLaMA-1-7B | 30.48 | 82.97 | - | - | - | - | 21.30 | 76.48 | - | - | - | - |
| BigTranslate-13B | 23.35 | 80.68 | 33.67 | 81.19 | 6.51 | 54.71 | 14.16 | 74.26 | 26.81 | 77.80 | 20.90 | 73.80 |
| Bayling-13B | 27.34 | 83.02 | 33.87 | 81.65 | - | - | 20.12 | 77.72 | 33.95 | 82.07 | - | - |
| *Our Recipe with Backbone Model: LLaMA-2-7B* | | | | | | | | | | | | |
| LLaMA-2-7B, zero-shot | 30.42 | 82.74 | 36.56 | 82.42 | 10.98 | 62.79 | 18.19 | 75.00 | 36.02 | 82.84 | 26.43 | 77.16 |
| ALMA-7B (Ours) | 29.49 | 83.98 | 42.91 | 85.90 | 35.26 | 85.97 | 23.52 | 79.73 | 38.93 | 84.81 | 34.02 | 84.08 |
| ALMA-7B-LoRA (Ours) | 29.56 | 83.95 | 43.49 | 85.93 | 35.64 | 86.09 | 23.64 | 79.78 | 39.21 | 84.84 | 34.31 | 84.12 |
| *Our Recipe with Backbone Model: LLaMA-2-13B* | | | | | | | | | | | | |
| LLaMA-2-13B, zero-shot | 31.06 | 83.01 | 40.02 | 83.27 | 15.77 | 66.35 | 21.81 | 78.10 | 36.50 | 82.91 | 29.03 | 78.73 |
| ALMA-13B (Ours) | 30.73 | 84.42 | 44.68 | 86.29 | 36.46 | 86.30 | 24.65 | 79.90 | 40.37 | 85.09 | 35.38 | 84.40 |
| ALMA-13B-LoRA (Ours) | **31.14** | **84.56** | **45.28** | **86.47** | **36.95** | **86.42** | **25.46** | **80.21** | 40.27 | **85.27** | **35.82** | **84.59** |

Table 2: The overall results in xx→en. ALMA models significantly outperform all prior similar studies and are comparable to SoTA models. The color and boldface are the same in Table 1.

**Compared with Prior Similar Studies** ALMA significantly outperforms all prior studies. Big-Translate, which is fine-tuned on Chinese corpus and 300M parallel corpus, struggles to surpass LLaMA-2's zero-shot performance, except for en→zh. This observation also aligns with our findings that an excessive amount of parallel data may damage the model, whereas target monolingual data is helpful to translation. Both TIM and SWIE specifically target two high-resource languages, de and zh. Their performance, however, is predominantly determined by their backbone models: effective translation is observed for zh but is lackluster for de when using BLOOMZ, and vice versa with LLaMA-1. In contrast, ALMA is versatile, showcasing strong results across all directions.

**Compared with SoTA models** Our best model (ALMA-13B-LoRA) substantially outperforms NLLB-54B and GPT-3.5-D on average. In en→xx direction, it even outperforms GPT-3.5-T on average COMET (87.00 vs. 86.56) and has close performance when it comes to xx→en. Notably, SoTA models typically excel with high-resource languages but falter with low-resource languages such as is. With our recipe, the performance of is remains strong and performs the best.

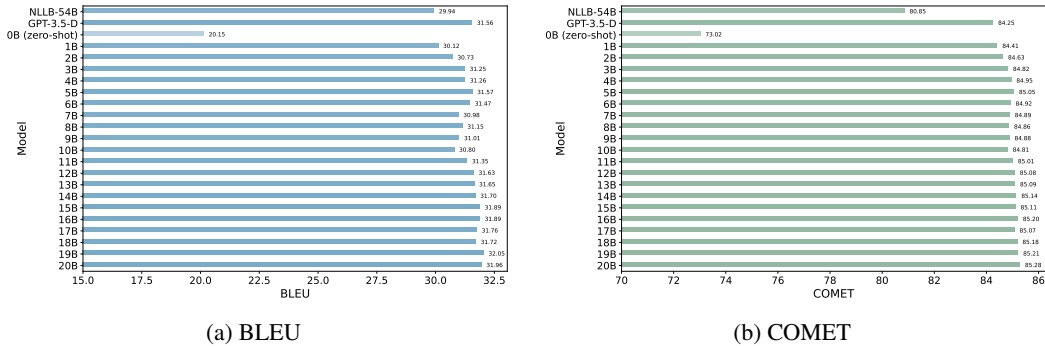

(a) BLEU

(b) COMET

Figure 5: The average performance of ALMA-7B at the completion of each 1B-token fine-tuning. The scores in the figure are averaged across 10 directions

# 6 ANALYSES

## 6.1 HOW MUCH MONOLINGUAL DATA TO USE?

In our main results, we present ALMA with our best settings, fine-tuned on either 20B or 12B tokens. Yet, we snapshot all ALMA models after every 1B monolingual tokens (and human-written parallel data) they have been fine-tuned with, and evaluate all their translation performance. As illustrated

| Use Mono. | Parallel Data Quality | Avg. xx→en | | Avg. en→xx | |
|---|---|---|---|---|---|
| | | BLEU | COMET | BLEU | COMET |
| ✘ | ✘ | 26.43 | 77.16 | 13.86 | 68.88 |
| ✘ | Random | 28.24 | 78.69 | 19.68 | 73.89 |
| ✘ | Filtered | 28.39 | 78.94 | 19.56 | 74.35 |
| ✘ | HW | 29.39 | 80.00 | 22.17 | 76.52 |
| ✔ | ✘ | 28.49 | 80.32 | 26.35 | 84.73 |
| ✔ | Random | 32.47 | 83.02 | 26.98 | 83.15 |
| ✔ | Filtered | 32.32 | 83.03 | 27.38 | 83.98 |
| ✔ | HW | **34.02** | **84.08** | **29.89** | **86.49** |

Table 3: Ablation study on the effect of monolingual data and parallel data quality. The backbone model is LLaMA-2-7B. A red cross (✘) in the table denotes the omission of monolingual data fine-tuning or parallel data (indicative of zero-shot translation). A green check (✔) signifies that the model undergoes fine-tuning with monolingual data.

in Figure 5, we report the ALMA-7B's average performance across all directions after fine-tuning every 1B tokens. The test dataset remains the same, i.e., the 10 aforementioned directions. We provide detailed numeric results and similar analysis for ALMA-13B to Appendix F. Importantly, merely fine-tuning on 1B monolingual tokens, followed by fine-tuning on human-written data, yields performance comparable to NLLB-54B and GPT-3.5-D. In practice, we employ 16 MI200 GPUs with a batch size of 256 and sequence length of 512, which requires only 18 hours to complete the fine-tuning of 1B tokens and an additional hour allocated for human-written data fine-tuning. It takes around 19 hours of training to have a strong MMT model.

## 6.2 The Effect of Monolingual Data and Parallel Data Quality

To scrutinize the impact of monolingual data, we juxtapose LLaMA-2-7B models fine-tuned with and without monolingual data (20B tokens), while keeping the same parallel data. Furthermore, to evaluate the impact of parallel data quality, we introduce three distinct parallel datasets for stage 2 fine-tuning. The first dataset is the human-written data (*HW*) utilized in prior experiments. The second is the filtered data (*Filtered*) referenced in Section 3.1. Lastly, we employ a randomly selected dataset (*Random*) sourced from the comprehensive WMT data. We anticipate the quality hierarchy as *HW*, followed by *Filtered*, and lastly, *Random*. For both *Filtered* and *Random*, each translation direction has 10K parallel data, aligning the total training dataset size with *HW*. We show the ablation results in Table 3. Using the LLaMA-2-7B as our foundational model, it's evident that with the same parallel data, incorporation of monolingual data largely enhances translation results, e.g., an increase from 74.35 to 83.98 in en→xx COMET scores when training on the same *Filtered* data. Moreover, regardless of the monolingual data's presence, models fine-tuned with higher-quality data exhibit better performance. Both monolingual and human-written data emerge as critical factors in improving translation. Detailed results for each language pair are deferred to the Appendix G.

## 6.3 Other Analyses

We also explore additional in-depth analyses and elaborate on them in the appendix: 1) The impact of the volume and domain of human-written data on translation performance is explored in Appendix H; 2) A comparison between stage 2 fine-tuning (parallel data fine-tuning) and in-context few-shot learning can be found in Appendix I; 3) An evaluation of the zero-shot cross-lingual capabilities of LLaMA-2 after stage 1 fine-tuning on other tasks is presented in Appendix J.

## 7 Conclusion

In this paper, we show that LLMs do not require as extensive a collection of parallel data as traditional translation models do. Subsequently, we introduce a novel training recipe for decoder-only LLMs in translation, resulting in strong translation models, ALMA. When using our LLaMA-2 as our foundational model, ALMA exceeds the zero-shot translation performance of LLaMA-2 by more than 12 BLEU and COMET scores across 10 directions on average. Moreover, ALMA models surpass all preceding studies and even outperform NLLB-54B and GPT-3.5-D.

ACKNOWLEDGMENTS

We extend our gratitude to Hieu Hoang, Marcin Junczys-Dowmunt, Yunmo Chen, Steven Tan, Huda Khayrallah, Thamme Gowda, Vikas Raunak, Matt Post, Anoop Kunchukuttan, Roman Grundkiewicz, Tom Kocmi, Kenton Murray and Arul Menezes for their insightful and valuable suggestions.

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

# A    COMPARING LLM TRAINING OBJECTIVES FOR MACHINE TRANSLATION

We evaluate three potential training objectives for decoder-only LLM in machine translation.

**Causal Language Modeling (CLM)**    We first consider a standard language modeling loss that predicts the next token based on all prior tokens.

**Prefix Language Modeling (Prefix LM)**    For decoder-only models, a prefix can be defined with a non-causal attention mask. Analogous to standard language modeling, the model predicts each token outside the prefix based on previous tokens. In the context of machine translation, the provided prompt serves as the prefix, as depicted in Figure 2.

**Mixture-of-Denoisers (MoD)**    The UL2 model (Tay et al., 2022a) introduces a unified approach to masking methods, utilizing a mixture-of-denoisers (MoD) strategy, which has also been implemented in the fine-tuning of PaLM (Tay et al., 2022b). This strategy is grounded in three objectives:

- *Regular Denoising*: In this approach, noise is sampled in spans and replaced with sentinel tokens, aligning with the standard span corruption technique delineated in Raffel et al. (2020). The parameters set for this objective include a mean of 3 and a corruption rate of 15
- *Extreme Denoising*: This method amplifies the noise to a comparatively 'extreme' level, characterized by a mean length of 32 and a corruption rate reaching up to 25
- *Sequential Denoising*: This is known as the Prefix LM objective previously mentioned.

In our training process, we allocate a 25% probability each for both regular and extreme denoising, and a 50% probability for sequential denoising.

We employ the MPT-7B as our backbone model. Our investigation considers four distinct training data sizes: 0 (zero-shot), 100K, 1M, and 5M, with translation directed from Russian to English. We use the parallel dataset previously described in Section 3.1. For each data size, the MPT-7B is fine-tuned using the corresponding training objective, noting that all trainings utilize full-weight fine-tuning.

The results of the comparison between training objectives can be viewed in Figure 6. Although three objectives end up with similar performance under 5M training data, both prefix LM and MoD markedly lag behind CLM under limited parallel data (100K or 1M). Surprisingly, with 100K, models fine-tuned using prefix LM and MoD even underperform their zero-shot performance. Conversely, CLM demonstrates a healthy improvement as the amount of parallel data increases. Consequently, we adopt CLM as our primary training objective for machine translation.

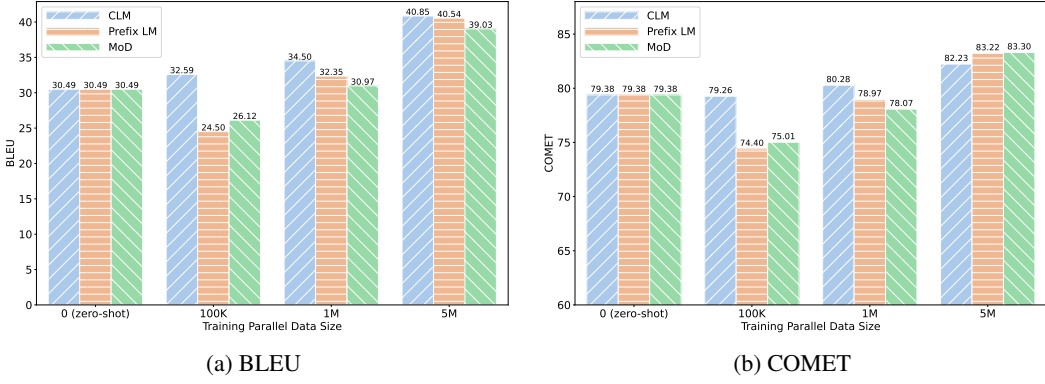

(a) BLEU                    (b) COMET

Figure 6: The comparison of translation performance across various training objectives and parallel data sizes is depicted. or datasets of 100K and 1M, both prefix LM and MoD lag behind CLM and even undeperform the zero-shot performance. Notably, only CLM demonstrates a healthy improvement as the volume of training data increases.

## B    FULL RESULTS OF ZERO-SHOT EVALUATION

In Section 2.2, we present the average zero-shot translation performance of recently released LLMs. Detailed results for each translation direction can be found in Table 4.

| Models | de | | cs | | is | | zh | | ru | | Avg. | |
|---|---|---|---|---|---|---|---|---|---|---|---|---|
| | BLEU | COMET | BLEU | COMET | BLEU | COMET | BLEU | COMET | BLEU | COMET | BLEU | COMET |
| *Translating from English* (en→xx) | | | | | | | | | | | | |
| OPT-7B | 9.79 | 65.74 | 2.95 | 51.55 | 1.42 | 45.66 | 1.59 | 48.84 | 1.31 | 41.57 | 3.41 | 50.67 |
| BLOOM-7B | 7.31 | 62.21 | 3.09 | 56.22 | 1.49 | **49.97** | 20.41 | 74.03 | 5.89 | 56.55 | 7.64 | 59.80 |
| Faclon-7B | 19.23 | 77.30 | 5.86 | 57.04 | 1.69 | 37.53 | **26.90** | 79.28 | 4.61 | 53.55 | 11.66 | 60.94 |
| LLaMA-1-7B | **21.00** | **79.50** | **16.31** | 78.16 | 2.42 | 34.92 | 15.63 | 68.03 | **17.61** | **76.73** | 14.59 | 67.47 |
| MPT-7B | 20.91 | 78.56 | 11.95 | 69.80 | **3.21** | 41.71 | 25.41 | **80.20** | 13.99 | 72.43 | **15.09** | 68.54 |
| LLaMA-2-7B | 19.00 | 76.39 | 16.02 | **79.13** | 1.33 | 43.83 | 16.97 | 71.80 | 16.00 | 73.24 | 13.86 | **68.88** |
| *Translating to English* (xx→en) | | | | | | | | | | | | |
| OPT-7B | 24.43 | 78.37 | 14.82 | 66.86 | 3.13 | 52.63 | 3.35 | 54.34 | 4.47 | 53.30 | 10.04 | 61.10 |
| BLOOM-7B | 22.06 | 74.10 | 6.06 | 55.18 | 2.14 | 48.70 | 13.66 | 74.62 | 20.06 | 69.27 | 12.80 | 64.37 |
| Faclon-7B | 29.21 | 82.02 | 20.06 | 71.15 | 4.29 | 52.53 | 19.45 | 76.68 | 19.50 | 73.19 | 18.50 | 71.11 |
| LLaMA-1-7B | 29.14 | 81.90 | 32.93 | 81.18 | 6.78 | 58.15 | 13.29 | 72.09 | 32.93 | 81.71 | 23.01 | 75.01 |
| MPT-7B | 29.32 | 81.80 | 27.45 | 76.12 | **12.44** | 62.76 | **19.72** | **77.25** | 30.55 | 79.21 | 23.90 | **75.43** |
| LLaMA-2-7B | **30.42** | **82.74** | **36.56** | **82.42** | 10.98 | **62.79** | 18.19 | 75.00 | **36.02** | **82.84** | **26.43** | 77.16 |

Table 4: The detailed results of LLM zero-shot performance in Figure 3

## C    TRAINING DETAILS

We fine-tune the backbone model using a warm-up ratio of 0.01, a maximum sequence length of 512 tokens, and a weight decay of 0.01. The test data from WMT'21 serves as our development set. The training spans 3 epochs (for MPT-7B as detailed in Section 3, and 2 epochs for LLaMA-2 human-written data fine-tuning). The best model is selected based on the lowest validation loss, with validation performed every 10% of the total training progress. We utilize 16 MI200 GPUs for training; each GPU manages 4 batches and has a gradient accumulation step of 4, yielding an effective batch size of 256. The peak learning rate is set at 2e-5 , with an inverse square learning rate decay to 0. The training operates under `fp16` precision, facilitated by deepspeed Rasley et al. (2020), employing ZeRO stage 2.

## D    DATA INFORMATION

### D.1    SAMPLING RATIO FOR MONOLINGUAL DATA

In Table 5, we observe a substantial imbalance in the volume of monolingual data available for different languages, denoted by their respective word counts[8]. Specifically, the English language dataset contains 523.9B words, vastly outnumbering other languages, such as Icelandic, which contains 0.3B words. Utilizing an unmodified concatenation and shuffling approach for this data would disproportionately prioritize English, undermining our objective of enhancing the model's proficiency in non-English languages. To address this, we straightforwardly set the sampling ratio for English as $P(l = \text{en}) = \frac{1}{6}$, thereby ensuring a balanced learning emphasis. The remaining $\frac{5}{6}$ of the probability allocation employs temperature sampling, as suggested by Aharoni et al. (2019), a technique prevalently adopted in the processing of unbalanced multilingual machine translation. Consequently, the process of selecting a monolingual example from language $l$ adheres to the following distribution:

$$P(l) \propto \left(\frac{D_l}{\sum_{l' \in L} D_{l'}}\right)^{\frac{1}{T}} \quad \text{s.t.} \quad \sum_{l' \in L} P(l') = \frac{5}{6} \tag{3}$$

where $D_l$ is the amount of the data in language $l$, $T$ is the temperature, and $L$ is the set of all languages except for English. The temperature we use is 6.

---

[8]https://huggingface.co/datasets/oscar-corpus/OSCAR-2301

| | Parallel Data | | | | Monolingual Data | |
|---|---|---|---|---|---|---|
| | Train | Development | Test (from English) | Test (to English) | # Words | Sampling Ratio |
| German (de) | 14211 | 1002 | 2037 | 1984 | 73.8B | 20% |
| Czech (cs) | 12076 | 1002 | 2037 | 1448 | 9.7B | 14% |
| Icelandic (is) | 2009 | - | 1000 | 1000 | 0.3B | 8% |
| Chinese (zh) | 15406 | 1002 | 2037 | 1875 | 44.4B | 19% |
| Russian (ru) | 15000 | 1002 | 2037 | 2016 | 78.0B | 22% |
| English (en) | - | - | - | - | 523.9B | 17% |

Table 5: The statistics for the data we utilize for the monolingual data fine-tuning and human-written data fine-tuning.

## D.2 DATA STATISTICS

We show data statistics in Table 5. The training parallel data is sourced from the WMT'17 to WMT'20. The development data was acquired from WMT'21, and the test data was derived from WMT'22, with the exception of the Icelandic dataset, which was procured from WMT'21. This means, Icelandic does not have development dataset. Additionally, the monolingual data was extracted from the Oscar dataset.

## E   OFF-TARGET ISSUE FOR LLaMA-2-13B

In the zero-shot scenario, the performance of LLaMA-2-13 is reasonable for translations into English. However, we identify a significant off-target issue with LLaMA-2-13B when translating from English to other languages. This issue is highlighted in Table 6 using a red highlighted box . An illustrative example of the off-target issue is provided below:

> *Translate this from English to Russian:*
> *English: Plug the wall charger (not included) to a power outlet, and then connect your eReader to the wall charger.*
> *Russian: Comment: I'm voting to close this question as off-topic because it is not about programming.*

| Models | de | | cs | | is | | zh | | ru | | Avg. | |
|---|---|---|---|---|---|---|---|---|---|---|---|---|
| | BLEU | COMET | BLEU | COMET | BLEU | COMET | BLEU | COMET | BLEU | COMET | BLEU | COMET |
| *Backbone Model: LLaMA-2-13B, Translating from English (en→xx)* | | | | | | | | | | | | |
| zero-shot | 13.69 | 75.55 | 0.87 | 68.57 | 2.36 | 38.47 | 30.00 | 79.70 | 0.59 | 63.84 | 9.50 | 65.23 |
| Prompt in Target Language | 25.91 | 81.88 | 20.80 | 81.82 | 2.05 | 40.80 | 31.82 | 82.08 | 22.66 | 83.29 | 20.65 | 73.97 |
| Filtered 1-shot | 25.71 | 80.85 | 20.77 | 81.30 | 2.78 | 42.97 | 31.70 | 82.12 | 22.32 | 83.03 | 20.66 | 74.05 |
| Filtered 5-shot | 26.32 | 81.67 | 20.89 | 81.45 | 2.78 | 42.62 | 32.01 | 82.02 | 23.26 | 83.28 | 21.05 | 74.21 |
| HW 5-shot | 26.33 | 82.63 | 21.87 | 82.66 | 3.04 | 41.93 | 30.73 | 82.65 | 22.77 | 84.21 | 20.95 | 74.82 |
| ALMA-13B-LoRA (ours) | **31.47** | **85.62** | **32.38** | **89.79** | **26.68** | **86.08** | **39.84** | **85.96** | **28.96** | **87.53** | **31.87** | **87.00** |
| *Backbone Model: LLaMA-2-13B, Translating to English (xx→en)* | | | | | | | | | | | | |
| zero-shot | 31.06 | 83.01 | 40.02 | 83.27 | 15.77 | 66.35 | 21.81 | 78.10 | 36.50 | 82.91 | 29.03 | 78.73 |
| Prompt in Target Language | 31.06 | 83.01 | 40.02 | 83.27 | 15.77 | 66.35 | 21.81 | 78.10 | 36.50 | 82.91 | 29.03 | 78.73 |
| Filtered 1-shot | 30.75 | 82.91 | 39.47 | 82.90 | 13.71 | 64.73 | 21.00 | 78.35 | 37.13 | 82.85 | 28.41 | 78.35 |
| Filtered 5-shot | 30.92 | 83.41 | 41.44 | 83.81 | 17.85 | 68.22 | 19.86 | 78.15 | 36.46 | 82.26 | 29.31 | 79.17 |
| HW 5-shot | 31.52 | 83.57 | 42.10 | 84.69 | 17.88 | 69.93 | 23.26 | 79.36 | 37.42 | 84.12 | 30.44 | 80.33 |
| ALMA-13B-LoRA (ours) | **31.14** | **84.56** | **45.28** | **86.47** | **36.95** | **86.42** | **25.46** | **80.21** | **40.27** | **85.27** | **35.82** | **84.59** |

Table 6: We demonstrate the off-target problem encountered during zero-shot translation from English to other languages using the LLaMA-2-13B model. Instances of this issue are highlighted within red boxes . Implementing prompts in the target languages and incorporating few-shot learning can markedly alleviate this issue. It is pertinent to note that the quality of the shots also influences the final outcomes.

Expectedly, the model should produce translations in Russian. Yet, LLaMA-2-13B outputs "I'm voting to ...", indicating a misinterpretation of the task, potentially linked to its pre-training phase. We address this off-target behavior through two methods.

**Prompt in the Target Language**   One approach is to utilize prompts in the target language (Raunak et al., 2023). For instance, when translating from English to Chinese, the preferred prompt

is: "将其从英文翻译成中文：\n英文：<source sentence>\n中文：" as opposed to "Translate this from English to Chinese:\nEnglish:<source sentence>\nChinese:". Employing this technique markedly enhances the zero-shot performance of LLaMA-2-13B. Specifically, the BLEU score escalates from 0.87 to 20.80 for en→cs, and from 0.59 to 22.66 for en→ru.

**In-Context Few-Shot Learning** Employing in-context few-shot learning by including several examples within the prompt has proven effective. We investigate both 1-shot and 5-shot learning scenarios. As delineated in Section I, we utilize two sets of examples: *Filtered*, extracted from the WMT training data, and another set randomly chosen from human-written data, termed *HW*. Table 6 demonstrates that both 1-shot and 5-shot configurations effectively counteract the off-target challenges. Few-shot learning exhibits performance comparable to the strategy of using prompts in the target language. Moreover, echoing observations from Section I, examples of human-written quality outperform those from the *Filtered* set.

Nevertheless, both strategies trail behind our proposed solution by a margin of approximately 5 BLEU and COMET points during translations into English, and by over 10 BLEU and COMET points in translations originating from English.

| Models | de | | cs | | is | | zh | | ru | | Avg. | |
|---|---|---|---|---|---|---|---|---|---|---|---|---|
| | BLEU | COMET | BLEU | COMET | BLEU | COMET | BLEU | COMET | BLEU | COMET | BLEU | COMET |
| *Translating from English* (en→xx) | | | | | | | | | | | | |
| NLLB-54B | **34.50** | **86.45** | **37.60** | **90.15** | 24.15 | 81.76 | 27.38 | 78.91 | **30.96** | **87.92** | 30.92 | 85.04 |
| GPT-3.5-D | 31.80 | 85.61 | 31.30 | 88.57 | 15.90 | 76.28 | 38.30 | **85.76** | 27.50 | 86.74 | 28.96 | 84.59 |
| 1B | 28.02 | 84.24 | 25.40 | 87.34 | 21.35 | 83.05 | 35.54 | 84.80 | 25.48 | 86.31 | 27.16 | 85.15 |
| 2B | 29.68 | 85.04 | 27.18 | 88.00 | 23.49 | 84.30 | 36.12 | 85.10 | 26.17 | 86.56 | 28.53 | 85.80 |
| 3B | 29.25 | 84.82 | 28.26 | 88.31 | 23.60 | 84.62 | 37.06 | 85.27 | 26.38 | 86.72 | 28.91 | 85.95 |
| 4B | 29.61 | 85.24 | 28.27 | 88.29 | 23.90 | 84.42 | 37.26 | 85.40 | 27.02 | 86.91 | 29.21 | 86.05 |
| 5B | 29.52 | 85.04 | 28.29 | 88.43 | 23.85 | 84.58 | 37.19 | 85.42 | 26.50 | 86.85 | 29.07 | 86.06 |
| 6B | 29.49 | 85.01 | 28.45 | 88.43 | 24.31 | 84.63 | 37.16 | 85.45 | 26.92 | 86.90 | 29.27 | 86.08 |
| 7B | 29.46 | 85.11 | 28.25 | 88.45 | 24.27 | 84.78 | 37.16 | 85.43 | 26.80 | 86.95 | 29.21 | 86.14 |
| 8B | 29.31 | 84.92 | 27.93 | 88.33 | 23.84 | 84.74 | 37.19 | 85.30 | 26.27 | 86.76 | 28.91 | 86.01 |
| 9B | 29.36 | 84.85 | 27.86 | 88.11 | 24.43 | 84.60 | 37.15 | 85.30 | 26.41 | 86.52 | 29.04 | 85.88 |
| 10B | 29.47 | 84.82 | 29.18 | 88.41 | 25.59 | 85.09 | 37.41 | 85.31 | 27.71 | 86.98 | 29.87 | 86.12 |
| 11B | 29.55 | 85.14 | 28.94 | 88.41 | 25.38 | 85.18 | 37.60 | 85.43 | 27.32 | 86.96 | 29.76 | 86.22 |
| 12B | 29.71 | 85.02 | 28.78 | 88.49 | 25.10 | 84.98 | 37.75 | 85.47 | 27.64 | 86.99 | 29.80 | 86.19 |
| 12B, *beam size=5* | 31.37 | 85.45 | 31.12 | 89.42 | **26.67** | **85.85** | **39.05** | 85.76 | 28.76 | 87.50 | **31.39** | **86.80** |
| *Translating to English* (xx→en) | | | | | | | | | | | | |
| NLLB-54B | 26.89 | 78.94 | 39.11 | 80.13 | 23.09 | 71.66 | 16.56 | 70.70 | 39.11 | 81.88 | 28.95 | 76.66 |
| GPT-3.5-D | **30.90** | **84.79** | 44.50 | 86.16 | 31.90 | 82.13 | **25.00** | **81.62** | 38.50 | 84.80 | 34.16 | 83.90 |
| 1B | 30.66 | 84.36 | 43.71 | 86.06 | 34.96 | 85.54 | 23.22 | 79.88 | 38.87 | 84.88 | 34.28 | 84.14 |
| 2B | 30.26 | 84.32 | 42.46 | 85.86 | 34.30 | 85.63 | 22.66 | 79.88 | 37.30 | 84.70 | 33.40 | 84.08 |
| 3B | 30.14 | 84.27 | 42.22 | 85.98 | 34.55 | 85.79 | 22.56 | 79.64 | 38.31 | 84.77 | 33.56 | 84.09 |
| 4B | 30.14 | 84.38 | 42.84 | 86.03 | 34.86 | 85.75 | 23.18 | 79.95 | 38.45 | 84.90 | 33.89 | 84.20 |
| 5B | 30.20 | 84.42 | 42.89 | 86.14 | 34.52 | 85.87 | 23.32 | 80.07 | 38.07 | 85.02 | 33.80 | 84.30 |
| 6B | 30.22 | 84.35 | 42.85 | 86.22 | 34.75 | 85.96 | 23.40 | 79.94 | 38.25 | 84.90 | 33.89 | 84.27 |
| 7B | 30.37 | 84.36 | 42.77 | 86.11 | 35.86 | 86.12 | 22.76 | 79.86 | 37.95 | 84.90 | 33.94 | 84.27 |
| 8B | 30.16 | 84.33 | 43.25 | 85.98 | 34.85 | 85.83 | 22.90 | 79.82 | 37.42 | 84.84 | 33.72 | 84.16 |
| 9B | 30.11 | 84.30 | 42.90 | 85.97 | 35.21 | 85.85 | 22.50 | 79.52 | 37.74 | 84.92 | 33.69 | 84.11 |
| 10B | 29.93 | 84.32 | 43.02 | 86.10 | 35.98 | 86.09 | 22.54 | 79.77 | 37.86 | 84.88 | 33.87 | 84.23 |
| 11B | 30.57 | 84.33 | 43.42 | 86.11 | 36.19 | 86.14 | 22.98 | 79.84 | 38.40 | 84.88 | 34.31 | 84.26 |
| 12B | 30.40 | 84.30 | 43.16 | 86.17 | 35.73 | 86.19 | 23.89 | 80.17 | 38.49 | 84.89 | 34.33 | 84.34 |
| 12B, *beam size=5* | 30.73 | 84.42 | **44.68** | **86.29** | 36.46 | 86.30 | 24.65 | 79.90 | **40.37** | **85.09** | 35.38 | 84.40 |

Table 7: The comprehensive numeric results for LLaMA-2-13B fine-tuned by every 1B monolingual tokens followed by human-written data fine-tuning.

# F NUMERIC RESULTS FOR MODELS FINE-TUNED WITH EVERY 1B TOKENS

In Table 7 and 8, results for LLaMA-2-13B and LLaMA-2-7B are presented. Both models were fine-tuned at every 1B-token interval (comprising six languages) before subsequent fine-tuning with human-written parallel data. Full-weight fine-tuning was employed to ensure a consistent comparison. During inference, the 7B models utilized a beam search of size 5, while the 13B models adopted a greedy search strategy. For 13B models, we only utilize a beam size 5 for the final models we reported in the main manuscript (Table 1 and 2).

The data from these tables highlight that fine-tuning only 1B tokens, followed by human-written data fine-tuning, is adequate to compete with or even outperform the state-of-the-art (SoTA) models.

| Models | de | | cs | | is | | zh | | ru | | Avg. | |
|---|---|---|---|---|---|---|---|---|---|---|---|---|
| | BLEU | COMET | BLEU | COMET | BLEU | COMET | BLEU | COMET | BLEU | COMET | BLEU | COMET |
| *Translating from English* (en→xx) | | | | | | | | | | | | |
| NLLB-54B | **34.50** | **86.45** | **37.60** | **90.15** | 24.15 | 81.76 | 27.38 | **78.91** | 30.96 | **87.92** | **30.92** | 85.04 |
| GPT-3.5-D | 31.80 | 85.61 | 31.30 | 88.57 | 15.90 | **76.28** | **38.30** | **85.76** | 27.50 | 86.74 | 28.96 | 84.59 |
| 1B | 28.40 | 84.45 | 26.99 | 87.91 | 20.64 | 83.22 | 35.09 | 84.41 | 25.10 | 86.33 | 27.24 | 85.26 |
| 2B | 28.96 | 84.62 | 28.05 | 88.34 | 22.23 | 84.44 | 34.39 | 84.08 | 26.02 | 86.37 | 27.93 | 85.57 |
| 3B | 29.10 | 84.66 | 28.68 | 88.46 | 23.23 | 84.74 | 35.50 | 84.40 | 26.35 | 86.75 | 28.57 | 85.80 |
| 4B | 29.02 | 84.75 | 28.14 | 88.53 | 23.78 | 84.94 | 35.51 | 84.65 | 26.22 | 86.68 | 28.53 | 85.91 |
| 5B | 29.34 | 84.89 | 29.00 | 88.82 | 24.16 | 84.76 | 35.82 | 84.71 | 26.21 | 86.74 | 28.91 | 85.98 |
| 6B | 28.78 | 84.61 | 28.31 | 88.56 | 23.85 | 84.82 | 34.96 | 84.43 | 26.03 | 86.67 | 28.39 | 85.82 |
| 7B | 28.72 | 84.83 | 27.72 | 88.49 | 23.88 | 84.86 | 35.18 | 84.33 | 26.17 | 86.54 | 28.33 | 85.81 |
| 8B | 29.03 | 84.78 | 28.76 | 88.64 | 23.49 | 84.94 | 35.38 | 84.66 | 26.42 | 86.45 | 28.62 | 85.89 |
| 9B | 28.97 | 84.79 | 28.06 | 88.39 | 23.57 | 85.04 | 35.11 | 84.49 | 26.20 | 86.70 | 28.38 | 85.88 |
| 10B | 29.25 | 84.81 | 27.97 | 88.52 | 23.55 | 85.08 | 35.60 | 84.66 | 26.18 | 86.58 | 28.51 | 85.93 |
| 11B | 29.62 | 85.23 | 28.77 | 88.68 | 24.27 | 85.08 | 35.75 | 84.73 | 26.55 | 86.91 | 28.99 | 86.13 |
| 12B | 29.85 | 85.15 | 28.90 | 88.67 | 24.68 | 85.27 | 36.31 | 84.78 | 26.95 | 87.00 | 29.34 | 86.17 |
| 13B | 29.88 | 85.20 | 29.30 | 88.80 | 24.78 | 85.24 | 36.35 | 84.77 | 26.98 | 87.05 | 29.46 | 86.21 |
| 14B | 29.95 | 85.23 | 29.59 | 89.09 | 25.02 | 85.20 | 36.37 | 84.83 | 27.00 | 87.10 | 29.59 | 86.29 |
| 15B | 30.10 | 85.22 | 29.79 | 89.09 | 25.21 | 85.40 | 36.27 | 84.78 | 27.37 | 86.94 | 29.75 | 86.29 |
| 16B | 30.12 | 85.32 | 29.65 | 89.14 | 24.87 | 85.34 | 36.58 | 84.93 | 26.97 | 86.98 | 29.64 | 86.34 |
| 17B | 30.07 | 85.32 | 29.32 | 88.71 | 25.28 | 85.13 | 36.24 | 84.89 | 27.43 | 87.05 | 29.67 | 86.22 |
| 18B | 29.63 | 85.40 | 29.14 | 89.02 | 25.11 | 85.33 | 36.64 | 84.96 | 26.96 | 87.02 | 29.50 | 86.35 |
| 19B | 30.01 | 85.25 | 29.75 | 89.06 | 25.66 | 85.37 | 36.87 | 85.11 | 27.13 | 86.98 | 29.88 | 86.35 |
| 20B | 30.31 | 85.59 | 29.88 | **89.10** | **25.71** | **85.52** | 36.48 | 85.05 | 27.09 | 87.17 | **29.89** | **86.49** |
| *Translating to English* (xx→en) | | | | | | | | | | | | |
| NLLB-54B | 26.89 | 78.94 | 39.11 | 80.13 | 23.09 | 71.66 | 16.56 | 70.70 | 39.11 | 81.88 | 28.95 | 76.66 |
| GPT-3.5-D | **30.90** | **84.79** | **44.50** | **86.16** | 31.90 | 82.13 | **25.00** | **81.62** | 38.50 | 84.80 | 34.16 | 83.90 |
| 1B | 29.40 | 83.99 | 41.64 | 85.54 | 33.35 | 84.76 | 22.45 | 79.12 | 38.15 | 84.34 | 33.00 | 83.55 |
| 2B | 29.53 | 84.00 | 43.32 | 85.66 | 33.79 | 85.17 | 22.19 | 78.98 | 38.82 | 84.59 | 33.53 | 83.68 |
| 3B | 30.15 | 84.00 | 43.08 | 85.79 | 34.43 | 85.47 | 22.70 | 79.29 | 39.32 | 84.61 | 33.94 | 83.83 |
| 4B | 29.82 | 83.98 | 43.26 | 85.92 | 34.55 | 85.59 | 23.27 | 79.84 | 39.00 | 84.62 | 33.98 | 83.99 |
| 5B | 30.09 | 84.15 | 43.39 | 85.97 | 35.26 | 85.77 | 23.65 | 80.05 | 38.81 | 84.65 | 34.24 | **84.12** |
| 6B | 30.26 | 84.00 | 43.91 | 85.86 | 35.46 | 85.82 | 23.75 | 79.85 | **39.37** | 84.58 | **34.55** | 84.02 |
| 7B | 29.44 | 83.87 | 42.53 | 85.90 | 34.33 | 85.71 | 23.23 | 79.76 | 38.60 | 84.58 | 33.63 | 83.96 |
| 8B | 29.69 | 84.00 | 42.85 | 85.68 | 34.38 | 85.69 | 22.92 | 79.31 | 38.54 | 84.47 | 33.68 | 83.83 |
| 9B | 29.76 | 83.94 | 42.89 | 85.90 | 34.47 | 85.46 | 23.03 | 79.57 | 38.01 | 84.50 | 33.63 | 83.87 |
| 10B | 29.05 | 83.87 | 41.88 | 85.61 | 33.69 | 85.40 | 22.97 | 79.29 | 37.83 | 84.28 | 33.08 | 83.69 |
| 11B | 29.39 | 83.82 | 43.42 | 85.95 | 34.87 | 85.69 | 22.68 | 79.61 | 38.13 | 84.44 | 33.70 | 83.90 |
| 12B | 29.49 | 84.00 | 43.22 | 85.97 | 35.24 | 85.74 | 22.94 | 79.65 | 38.72 | 84.59 | 33.92 | 83.99 |
| 13B | 29.51 | 83.94 | 42.90 | 85.88 | 35.34 | 85.78 | 22.80 | 79.64 | 38.65 | 84.60 | 33.84 | 83.97 |
| 14B | 29.65 | 83.93 | 42.89 | 85.83 | 35.47 | 85.83 | 22.40 | 79.65 | 38.67 | 84.70 | 33.82 | 83.99 |
| 15B | 29.48 | 83.89 | 43.21 | 85.93 | 35.72 | 85.88 | 22.74 | 79.34 | 39.04 | 84.63 | 34.04 | 83.93 |
| 16B | 29.67 | 83.99 | 43.27 | 86.03 | 35.54 | 85.90 | 22.95 | 79.57 | 39.28 | 84.79 | 34.14 | 84.06 |
| 17B | 29.48 | 83.93 | 43.16 | 85.81 | 35.19 | 85.81 | 22.90 | 79.35 | 38.50 | 84.66 | 33.85 | 83.91 |
| 18B | 29.43 | 83.98 | 43.25 | 85.94 | 35.16 | 85.83 | 23.57 | 79.69 | 38.32 | 84.64 | 33.95 | 84.02 |
| 19B | 29.54 | 84.06 | 42.85 | 85.83 | **35.97** | **86.03** | 23.42 | 79.56 | 39.34 | **84.83** | 34.22 | 84.06 |
| 20B | 29.49 | 83.98 | 42.91 | 85.90 | 35.26 | 85.97 | 23.52 | 79.73 | 38.93 | 84.81 | 34.02 | 84.08 |

Table 8: The comprehensive numeric results for LLaMA-2-7B fine-tuned by every 1B monolingual tokens followed by human-written data fine-tuning.

## G   DETAILED RESULTS IN ABLATION STUDY

We show the detailed results of the ablation study on the effect of monolingual data and the quality of the data in Table 9.

## H   IS MORE HUMAN-WRITTEN PARALLEL DATA BETTER?

The composition of our human-written data consists of the prior-year WMT test sets (approximately 10K parallel sentences per pair) and Flores data (around 2K per pair). In this analysis, we assess the

| Use mono. | Parallel Data Quality | de | | cs | | is | | zh | | ru | | Avg. | |
|---|---|---|---|---|---|---|---|---|---|---|---|---|---|
| | | BLEU | COMET | BLEU | COMET | BLEU | COMET | BLEU | COMET | BLEU | COMET | BLEU | COMET |
| | | *Translating from English* (en→xx) | | | | | | | | | | | |
| ✗ | ✗ | 19.00 | 76.39 | 16.02 | 79.13 | 1.33 | 43.83 | 16.97 | 71.80 | 16.00 | 73.24 | 13.86 | 68.88 |
| ✗ | Random | 22.74 | 78.06 | 19.38 | 79.59 | 6.20 | 50.45 | 27.66 | 79.70 | 22.40 | 81.64 | 19.68 | 73.89 |
| ✗ | Filtered | 21.92 | 77.59 | 19.93 | 80.24 | 6.91 | 51.42 | 27.15 | 80.09 | 21.90 | 82.42 | 19.56 | 74.35 |
| ✗ | HW | 27.30 | 83.46 | 22.59 | 84.59 | 3.61 | 45.81 | 33.74 | 83.81 | 23.63 | 84.94 | 22.17 | 76.52 |
| ✔ | ✗ | 27.44 | 84.17 | 28.61 | 88.57 | 21.55 | 83.69 | 28.51 | 81.56 | 25.65 | 85.67 | 26.35 | 84.73 |
| ✔ | Random | 27.38 | 82.12 | 28.43 | 86.82 | 21.65 | 80.53 | 31.68 | 81.73 | 25.74 | 84.53 | 26.98 | 83.15 |
| ✔ | Filtered | 27.97 | 83.16 | 28.45 | 87.26 | 23.03 | 82.40 | 31.55 | 82.26 | 25.92 | 84.84 | 27.38 | 83.98 |
| ✔ | HW | **30.31** | **85.59** | **29.88** | **89.10** | **25.71** | **85.52** | **36.48** | **85.05** | **27.09** | **87.17** | **29.89** | **86.49** |
| | | *Translating to English* (xx→en) | | | | | | | | | | | |
| ✗ | ✗ | **30.42** | 82.74 | 36.56 | 82.42 | 10.98 | 62.79 | 18.19 | 75.00 | 36.02 | 82.84 | 26.43 | 77.16 |
| ✗ | Random | 29.15 | 82.33 | 38.61 | 82.67 | 17.14 | 68.25 | 19.32 | 77.24 | 36.98 | 82.97 | 28.24 | 78.69 |
| ✗ | Filtered | 29.29 | 82.42 | 38.41 | 82.80 | 17.89 | 69.05 | 19.22 | 77.41 | 37.12 | 83.04 | 28.39 | 78.94 |
| ✗ | HW | 29.95 | 83.93 | 40.32 | 84.31 | 15.61 | 69.13 | 22.51 | 78.77 | 38.56 | 83.88 | 29.39 | 80.00 |
| ✔ | ✗ | 28.28 | 82.48 | 38.05 | 84.18 | 32.79 | 84.07 | 9.44 | 69.71 | 33.88 | 81.18 | 28.49 | 80.32 |
| ✔ | Random | 28.89 | 82.74 | 40.64 | 85.01 | 35.11 | 85.67 | 19.50 | 77.67 | 38.19 | 84.01 | 32.47 | 83.02 |
| ✔ | Filtered | 28.63 | 82.85 | 40.93 | 84.85 | 35.12 | 85.49 | 19.04 | 77.92 | 37.90 | 84.02 | 32.32 | 83.03 |
| ✔ | HW | 29.49 | **83.98** | **42.91** | **85.90** | 35.26 | **85.97** | 23.52 | **79.73** | **38.93** | **84.81** | **34.02** | **84.08** |

Table 9: Detailed results of ablation study on the effect of monolingual data and parallel data quality. The backbone model is LLaMA-2-7B. A red cross (✗) in the table denotes the omission of monolingual data fine-tuning or parallel data (indicative of zero-shot translation). A green check (✔) signifies that the model undergoes fine-tuning with monolingual data.

| Parallel Data Used | Avg. xx→en | | Avg. en→xx | |
|---|---|---|---|---|
| | BLEU | COMET | BLEU | COMET |
| | *Backbone: LLaMA-2-7B After Stage 1* | | | |
| Flores | 30.50 | 83.24 | 29.28 | **86.52** |
| Flores+WMT | **34.02** | **84.08** | **29.89** | 86.49 |

Table 10: The performance of LLaMa-2-7B (post stage 1 fine-tuning) when fine-tuned exclusively on Flores versus when fine-tuned on both WMT and Flores.

impact of additional human-written parallel data. Specifically, we compare models (LLaMa-2-7B after stage 1) fine-tuned exclusively on Flores against those fine-tuned on both Flores and WMT data. Results can be found in Table 10. Notably, upon integrating WMT data into the training set, we discern a modest improvement in COMET scores. However, there's an uptick in BLEU scores, particularly for translations into English. We attribue the increase in lexical match (BLEU) to the domain alignment of WMT data. Consequently, our hypothesis is that while an augmented volume of human-written data might marginally enhance segment-level human judgment correlation (COMET), in-domain data can significantly enhance lexical matching.

| Methods | Avg. xx→en | | Avg. en→xx | |
|---|---|---|---|---|
| | BLEU | COMET | BLEU | COMET |
| | *Backbone: LLaMA-2-13B After Stage 1* | | | |
| Zero-Shot | 33.07 | 83.07 | 26.76 | 84.03 |
| Filtered 5-shot | 33.12 | 83.13 | 27.50 | 83.78 |
| HW 5-shot | 33.75 | 83.91 | 27.59 | 85.24 |
| Our Stage 2 | **34.31** | 84.12 | **29.78** | **86.37** |
| Our Stage 2 + HW 5-shot | 34.14 | **84.27** | 28.56 | 85.87 |

Table 11: The performance between 5-shot ICL and stage 2 fine-tuning using the LLaMA-2-13B model post stage 1 as the backbone. Our findings indicate that the quality of shots affects ICL performance. Notably, stage 2 fine-tuning markedly surpasses the 5-shot ICL and ICL does not help more on stage 2.

# I  PARALLEL DATA FINE-TUNING VS. IN-CONTEXT LEARNING

An alternative way to instruct the model to have better translation is in-context learning (ICL) (Brown et al., 2020), as opposed to additional fine-tuning on parallel data. However, ICL is limited to only a few shots given the length of translation examples, while fine-tuning can leverage

entirely available data. For ICL, we consider 5-shot evaluations. 5 examples are randomly selected from *Filtered* data (the *Quality-Random* examples used by Hendy et al. (2023)). We also consider another 5 examples randomly from the human-written data to examine the impact of example quality. We here compare the performance of our fine-tuning method and 5-shot ICL.[9] We assess the LLaMA-2-13B after stage 1 (12B token fine-tuning) and present results in Table 11.

Interestingly, ICL also holds the same property that higher quality data leads to better performance (Filtered 5-shot vs. HW 5-shot). Moreover, as expected, ICL substantially underperforms our stage 2 fine-tuning possibly due to the small examples provided, which aligns with the findings in the previous work (Liu et al., 2022; Mosbach et al., 2023). This could also clarify why implementing ICL subsequent to stage 2 yields no additional benefits, as all high-quality data has already been incorporated during stage 2 fine-tuning (the last row in the Table).

## J    CROSS-LINGUAL PROFICIENCY OF FINE-TUNED MODELS

We explore the cross-lingual competencies of our models derived from LLaMA-2 after fine-tuning them on monolingual data. Our aim is to discern whether augmenting monolingual data enhances performance in cross-lingual tasks. Experiments were conducted on zero-shot cross-lingual tasks encompassing three benchmarks: Cross-lingual language understanding (XNLI) Conneau et al. (2018), XStoryCloze—a translation of the English StoryCloze dataset into ten languages (Mostafazadeh et al., 2017), and XWinograd—a multilingual compilation of Winograd Schemas Tikhonov & Ryabinin (2021). Evaluations were restricted to languages overlapping with our fine-tuned languages, namely, German (with only XNLI being inclusive), Chinese, Russian, and English. Unfortunately, none of these datasets covers Icelandic. We first consider baselines for some widely used models: XLM-R large (Conneau et al., 2020), XGLM-7.5B (Lin et al., 2021), BLOOM-7B (Scao et al., 2022), and MPT-7B (MosaicML, 2023). In these comparisons, LLaMA-2 demonstrates the top performance for the tested languages. Subsequent fine-tuning with either 1B or 20B monolingual tokens on both LLaMA-2-7B and 13B models yields substantial enhancements for non-English languages across all tasks. A consistent trend observed was that increased monolingual data corresponds to greater performance boosts. Only English is observed for a negligible difference after fine-tuning monolingual data, which is an anticipated outcome given LLaMA-2's proficient grasp of English. The tool we utilize for LLM evaluation is `lm-evaluation-harness` (Gao et al., 2021).[10]

| Models | XNLI | | | | | Xstorycloze | | | | XWinograd | | | |
|---|---|---|---|---|---|---|---|---|---|---|---|---|---|
| | de | en | ru | zh | Avg. | en | ru | zh | Avg. | en | ru | zh | Avg. |
| XLMR-Large | 32.29 | 27.51 | 31.20 | 33.41 | 31.10 | 49.83 | 48.25 | 46.46 | 48.18 | 47.31 | 48.89 | 44.05 | 46.75 |
| XGLM-7.5B | 48.31 | 54.06 | 46.55 | 34.66 | 45.90 | 69.82 | 63.34 | 58.90 | 64.02 | 79.35 | 63.17 | 72.82 | 71.78 |
| BLOOM-7B | 39.04 | 53.37 | 42.61 | 35.50 | 42.63 | 70.48 | 52.68 | 61.88 | 61.68 | 82.06 | 56.83 | 74.21 | 71.03 |
| MPT-7B | 47.87 | 55.62 | 46.10 | 36.71 | 46.58 | 78.09 | 57.71 | 59.50 | 65.10 | 86.58 | 68.89 | 73.21 | 76.23 |
| LLaMA-2-7B | 45.90 | 56.51 | 41.33 | 34.82 | 44.64 | 77.04 | 63.07 | 59.56 | 66.56 | 87.91 | 68.89 | 70.63 | 75.81 |
| LLaMA-2-7B, 1B mono. | 47.63 | **57.87** | 44.02 | 34.70 | 46.06 | 75.84 | 64.59 | 60.03 | 66.82 | 85.63 | 66.98 | 71.83 | 74.81 |
| LLaMA-2-7B, 20B mono. | **50.48** | 57.35 | 46.47 | 33.82 | 47.03 | 75.71 | 67.57 | 62.81 | 68.70 | 86.06 | 66.67 | 75.20 | 75.98 |
| LLaMA-2-13B | 49.08 | 53.21 | 44.74 | 36.14 | 45.79 | **78.36** | 66.18 | 63.40 | 69.31 | **88.99** | 68.25 | 77.98 | 78.41 |
| LLaMA-2-13B, 1B mono. | 47.27 | 49.56 | 44.14 | 38.35 | 44.83 | 78.09 | 68.63 | 62.01 | 69.58 | 88.47 | 69.21 | **78.17** | 78.62 |
| LLaMA-2-13B, 12B mono. | 49.56 | 54.02 | **47.55** | **40.72** | **47.96** | 78.29 | **69.82** | **63.67** | **70.59** | 88.82 | **70.16** | **78.17** | **79.05** |

Table 12: We evaluate the zero-shot cross-lingual efficacy on three multilingual datasets. Our findings indicate that fine-tuning LLaMA-2 with more monolingual data results in enhanced performance for non-English languages.

---

[9]Due to ICL's extended prompt length, employing a large beam size is impractical; hence, we opt for a beam size of 1 for all to ensure a fair comparison.

[10]https://github.com/EleutherAI/lm-evaluation-harness/tree/big-refactor

