# OpenReview forum: "A Paradigm Shift in Machine Translation: Boosting Translation Performance of Large Language Models"
_ICLR.cc/2024/Conference — ICLR 2024 poster_

### Official Review · Reviewer_TMW7 · 2023-10-30

**Soundness:** 3 good
**Presentation:** 3 good
**Contribution:** 3 good
**Rating:** 8
**Confidence:** 4

**Summary:**

This paper proposed an innovative two-stage fine-tuning method: initially fine-tuning on non-English monolingual data to enhance comprehension, followed by further fine-tuning on a small amount of high-quality human-translated parallel text. This approach enabled even smaller LLMs to achieve state-of-the-art translation performance.

**Strengths:**

The results of the paper indicated that smaller models could achieve SOTA translation levels through specialized fine-tuning, suggesting that there might not be a continuous need to expand datasets and models for better performance.

Through compact, specialized fine-tuning, smaller LLMs could achieve SOTA translation quality without billions of parameters. The focus of this research was on tailored fine-tuning methods that unleashed the potential of LLM's multilingual capabilities on a broader scale.

The paper demonstrated that instead of increasing data scale, intentional fine-tuning targeting key language capabilities might be the key to maximizing LLM performance.

By revealing the potential of smaller LLMs for efficient and accurate machine translation, this work laid the foundation for developing more user-friendly and scalable machine translation systems. This training approach offered more possibilities for deploying capable multilingual LLMs in real-world applications.

**Weaknesses:**

There were certain flaws in the method, and prompts affected the results.

The evaluation methods had its limitations.

The stability of the proposed method was not verified.

**Questions:**

1. Typically, if an LLM was fine-tuned with specific bilingual corpora to enhance translation capabilities between those two languages, it might impair other NLP capabilities of the LLM, such as document summarization and logical question answering. Did this issue not arise in this study?

2. If possible, please consider using the 'Instruct Score' from EMNLP 2023 (Instructscore: Towards Explainable Text Generation Evaluation with Automatic Feedback) as a metric. I believe it's a better benchmark for evaluating LLM-MT.

3. The outputs of large models were uncertain. Even a minor change in a prompt could lead to variations in the output. During the two-stage fine-tuning process, was the specific impact of the prompt considered?

3. Given the powerful In-Context Learning capabilities of large language models, it would be worth exploring whether adding relevant knowledge to the prompt could further enhance translation capabilities.

4. The section "Small Training Data Is Enough" contained many uncertain descriptions, which should be rigorous in a paper and supported by convincing data. Moreover, the best-performing commercial LLM, GPT-4, remained proprietary, leaving us in the dark about the amount of data used, the number of parameters trained, and potential data leakage issues during translation metric testing.

5. The sentence in the "Large Parallel Data Wash Out the Knowledge" section: "As expected, it tends up with a similar performance in both BLEU and COMET evaluations (triangle in Figure 4)," was hard to understand.

6. The proposed method significantly improved translation metric scores, highlighting ALMA's effectiveness. Based on this, how did the authors believe LLM generalized translation capabilities? Did the proposed method fundamentally assist LLMs in learning deep bilingual text alignment?

---

> ### Author Response · Authors · 2023-11-17
> **Official Rebuttal 1 by Authors**
>
> We genuinely appreciate the valuable feedback provided by the reviewer and have addressed them in a point-by-point manner below. We are more than willing to engage in further discussions with the reviewers should any follow-up questions arise.
>
> **Regarding your concern about prompt & stability :**
> >There were certain flaws in the method, and prompts affected the results.
>
> >The stability of the proposed method was not verified.
>
> Thank you for your insightful feedback!  We wholeheartedly acknowledge the idea that different prompts can influence translation performance, and that enriching these prompts could further improve results. Our primary focus, however, is to demonstrate that a properly formulated prompt enables moderate-sized large language models (LLMs) to achieve performance comparable to state-of-the-art (SOTA) encoder-decoder translation models or GPT3.5/4. The prompts we have employed are widely recognized and validated in the field of machine translation, as evidenced by previous research ([1], [2], [3]). Nonetheless, we agree that exploring alternative prompts could potentially yield even better translation outcomes! Hence, we re-train ALMA-7B-LoRA model with **two new prompts** as suggested in [4]:
>
> Recap that:
> - Our prompt is: Translate this from <src> to <tgt>:\n<src>: <input>\n <tgt>:
> - New Prompt 1: <src>: <input>\n<tgt>:
> - New Prompt 2: <input> Translate this from <src> to <tgt>:
>
> |                | Avg. en>xx |       | Avg. xx>en |       |
> |----------------|------------|-------|------------|-------|
> | Prompt (for ALMA-7B-LoRA) | BLEU  | Comet22 | BLEU  | Comet22 |
> | Our prompt               | **29.78** | **86.80**   | **34.34** | **84.24**   |
> | New prompt 1             | 27.63 | 85.44   | 33.434| 83.60   |
> | New prompt 2             | 27.44 | 85.31   | 33.314| 83.85   |
>
> We conducted evaluations of the two new prompts using the WMT’22 test set and report the average scores for both translation directions. The performance of both new prompts is comparable, though they slightly underperform compared to our originally proposed prompt.
>
> Reference:
>
> [1] Hendy A, Abdelrehim M, Sharaf A, Raunak V, Gabr M, Matsushita H, Kim YJ, Afify M, Awadalla HH. How good are gpt models at machine translation? a comprehensive evaluation. arXiv preprint arXiv:2302.09210. 2023 Feb 18.
>
> [2] Chen Y, Liu Y, Meng F, Chen Y, Xu J, Zhou J. Improving Translation Faithfulness of Large Language Models via Augmenting Instructions. arXiv preprint arXiv:2308.12674. 2023 Aug 24.
>
> [3] Zeng J, Meng F, Yin Y, Zhou J. Tim: Teaching large language models to translate with comparison. arXiv preprint arXiv:2307.04408. 2023 Jul 10.
>
> [4] Zhang B, Haddow B, Birch A. Prompting large language model for machine translation: A case study. arXiv preprint arXiv:2301.07069. 2023 Jan 17.
>
> **Regarding your Concern about evaluation metric:**
> Thank you for your constructive feedback. We acknowledge that finding reliable evaluation metrics remains a challenging aspect of machine translation research. During our experiments, we reported results using COMET-22 and BLEU, which were the predominant metrics at that time. These metrics are not only widely accepted but also facilitate easy comparison within the research community. To address your concerns, we have also evaluated our ALMA models using three additional automatic metrics:
>
> Thank you for recommending InstructScore, which is an intriguing automatic evaluation metric. We observed that the prompt for this metric (found at https://github.com/xu1998hz/SEScore3/blob/586f7f22ced93b136cbde818e86fbddcdd8580ce/InstructScore.py#L67C19-L67C19) is specifically tailored for translations from Chinese to English. Therefore, we limited our evaluation of the Instructscore to this language pair. The results, as detailed in the table below, demonstrate that our method surpasses GPT-3.5-D in performance, with higher scores indicating better results.
>
> |             | zh->en (Instructscore) |
> |-------------|------------------------|
> | GPT-3.5-D   | -5.2944                |
> | ALMA-13B-LoRA | **-4.5856**              |
>
> To provide a more thorough assessment using recently released evaluation metrics, we conducted evaluations using wmt23-cometkiwi-da-xxl, and Unbabel/XCOMET-XXL. Both of these are 10 billion parameter models that support all languages. The results, as presented in the table below, indicate that our method continues to outperform GPT-3.5-D when evaluated with these latest metrics.
>
> |              | Avg. xx->en |       |            |
> |-|---|--|---|
> |              | Comet22     | wmt23-cometkiwi-da-xxl | XCOMET-XXL |
> | GPT-3.5-D    | 83.90       | 80.06 | 84.59      |
> | ALMA-13B-LoRA| **84.59**       | **81.50** | **86.74**      |
>
>
> |              | Avg. en->xx |       |            |
> |------|---|-------|------|
> |              | Comet22     | wmt23-cometkiwi-da-xxl | XCOMET-XXL |
> | GPT-3.5-D    | 84.59       | 75.99 | 88.10      |
> | ALMA-13B-LoRA| **87.00**       | **82.66** | **92.76**      |

---

> > ### Author Response · Authors · 2023-11-17
> > **Official Rebuttal 2 by Authors**
> >
> > **Regarding your question 1:**
> > >Typically, if an LLM was fine-tuned with specific bilingual corpora to enhance translation capabilities between those two languages, it might impair other NLP capabilities of the LLM, such as document summarization and logical question answering. Did this issue not arise in this study?
> >
> > Thank you for posing such an insightful question, which indeed addresses a key aspect of our research. You are likely aware that the first stage of our process, monolingual fine-tuning, is designed to enhance the large language model's (LLM) proficiency in general non-English languages. The second stage, involving parallel data fine-tuning, is specifically tailored for machine translation.
> >
> > In our study, we have confirmed that the first stage of fine-tuning does not significantly impact the model's performance in English across various tasks and highly improves for non-English languages, as shown in Table 12. For the second stage, we have explored the use of Low-Rank Adaptation (LoRA) during fine-tuning with bilingual corpora. This approach is intended to avoid any potential negative impact on other NLP capabilities after fine-tuning. Importantly, LoRA is utilized as a **lightweight, plug-and-play module**. This means that, based on the model fine-tuned in the first stage, we can extend its application to develop multilingual summarization or question-answering capabilities by training a corresponding LoRA module for each task. This approach ensures that our fine-tuning process enhances specific functionalities while preserving the model's overall proficiency in diverse NLP tasks.
> >
> > **Regarding your question 2:**
> > >If possible, please consider using the 'Instruct Score' from EMNLP 2023 (Instructscore: Towards Explainable Text Generation Evaluation with Automatic Feedback) as a metric. I believe it's a better benchmark for evaluating LLM-MT.
> >
> > Thank you for the question! Please refer to our answers in **“Regarding your Concern about evaluation metric”** above.
> >
> > **Regarding your question 3 & 4:**
> > >The outputs of large models were uncertain. Even a minor change in a prompt could lead to variations in the output. During the two-stage fine-tuning process, was the specific impact of the prompt considered?
> > Given the powerful In-Context Learning capabilities of large language models, it would be worth exploring whether adding relevant knowledge to the prompt could further enhance translation capabilities.
> >
> > Thanks for your questions! Please refer to our answers in **“Regarding your concern about prompt & stability”** above
> >
> > **Regarding your question 5:**
> > >The section "Small Training Data Is Enough" contained many uncertain descriptions, which should be rigorous in a paper and supported by convincing data. Moreover, the best-performing commercial LLM, GPT-4, remained proprietary, leaving us in the dark about the amount of data used, the number of parameters trained, and potential data leakage issues during translation metric testing.
> >
> > Thank you for your thorough examination of our claim in the "Small Training Data Is Enough" section. We appreciate the opportunity to engage in a more detailed discussion on this topic. In light of your feedback, we acknowledge that certain descriptions in this section, such as "well-trained LLM may not necessitate substantial parallel data," might be perceived as overly broad and lacking rigor. We intend to revise this to specifically reference "LLaMA-2" instead of a general "well-trained LLM."
> >
> > Additionally, we fully agree with your observation regarding the unknown aspects of GPT-4. This lack of transparency is precisely why our study is significant. Our goal is to develop and implement multilingual LLMs that are not only entirely transparent but also freely available for real-world applications. This approach is intended to provide a clear and open understanding of the LLM's functioning and training, contrasting with the more opaque nature of models like GPT-4.

---

> > > ### Author Response · Authors · 2023-11-17
> > > **Official Rebuttal 3 by Authors**
> > >
> > > **Regarding your question 6:**
> > > >The sentence in the "Large Parallel Data Wash Out the Knowledge" section: "As expected, it tends up with a similar performance in both BLEU and COMET evaluations (triangle in Figure 4)," was hard to understand.
> > >
> > > Thank you for your question, which invites a deeper exploration of our hypothesis. We propose that excessive parallel data (for instance, 20 million sentences) might dilute the pre-existing knowledge in pre-trained LLMs like LLaMA-2 and MPT. A pertinent question arises: To what extent does this "knowledge washout" occur?
> > >
> > > To investigate this, we conducted an experiment where we trained a model from scratch with 20 million sentences. This simulates an extreme scenario where all pre-trained knowledge is effectively erased, leaving the model with knowledge derived solely from the 20 million parallel sentences. The COMET score for this model, trained from scratch, was 77.5. This score is strikingly close to those of LLaMA-2 or MPT, which hover around 79. This similarity suggests that, with an increase in parallel data and extended training, all LLMs sharing a similar architecture (even those trained from scratch) might eventually converge towards similar translation models, effectively erasing their previous specialized knowledge.
> > >
> > > Therefore, we conclude that using a large volume of parallel data may not be the most appropriate approach for LLMs, as it can lead to the loss of their uniquely trained features and knowledge base.
> > >
> > > **Regarding your question 7**
> > > >The proposed method significantly improved translation metric scores, highlighting ALMA's effectiveness. Based on this, how did the authors believe LLM generalized translation capabilities?
> > >
> > > Thank you for your thought-provoking question and insightful feedback.
> > >
> > > To begin, let's consider the generalized translation capabilities of large language models (LLMs). LLMs acquire extensive general knowledge from the massive volumes of monolingual data they process, as seen in our first stage of fine-tuning. We posit that this is a significant advantage over traditional encoder-decoder models, which rely solely on parallel data for text alignment learning. LLMs inherently possess translation capabilities; our task in the second stage is to guide the model to effectively utilize this information and produce accurate outputs. This second stage aligns more closely with AI alignment goals, directing LLM systems to achieve the specific intentions of the designer—in this case, translation. The ability of LLMs to generalize translation capabilities is attributed to their exposure to a vast array of monolingual data. For example, let's consider an example involving specific Chinese internet slang. The terms “八卦” (bā guà) and “实锤” (shí chuí) literally translate to “eight trigrams" and "solid hammer" respectively, but in slang, they mean “gossip" and "solid truth." When translating the phrase “这个八卦实锤了”, most translation systems, including NLLB-54B, face challenges. However, our ALMA-13B-LoRA model exhibits a more natural and accurate understanding in its translation:
> > >
> > > - Source Chinese sentence: "这个八卦实锤了"
> > > - NLLB-54B translation: "This is a real shame."
> > > - ALMA-13B-LoRA translation: "The gossip is true."
> > >
> > > Moreover, our results in Table 10 show that even when our training data is limited to out-of-domain Flores data, the COMET score remains high when tested on the cross-domain WMT’22 dataset, which includes conversational, news, e-commerce, and social content. This further validates the generalization capability of ALMA models.
> > >
> > > > Did the proposed method fundamentally assist LLMs in learning deep bilingual text alignment?
> > >
> > > As for the question of whether our method fundamentally assists LLMs in learning deep bilingual text alignment, we wouldn't claim it to be "fundamental" at this stage. We are still investigating the optimal approach to aid LLMs in achieving satisfactory translation. However, we believe that simple supervised fine-tuning on high-quality but parallel data is a viable method for achieving strong text alignment. Future research is likely to yield even more effective strategies.

---

> > > > ### Comment · Reviewer_TMW7 · 2023-11-22
> > > >
> > > > Thanks to the authors for their patient responses and the extensive experimental validations provided for the questions raised. Although the responses are somewhat cautious, they adequately address the issues raised.
> > > >
> > > > I have revised my rating to an 8, and I believe the paper is worthy of acceptance.

---

> > > > > ### Author Response · Authors · 2023-11-23
> > > > >
> > > > > Thank you! Your detailed and insightful comments are greatly appreciated. They are crucial in enhancing the quality of our work.

---

### Official Review · Reviewer_AH62 · 2023-10-31

**Soundness:** 3 good
**Presentation:** 2 fair
**Contribution:** 2 fair
**Rating:** 5
**Confidence:** 4

**Summary:**

This paper examines ways of improving the translation performance of large language models. There is no modelling or training innovation in the paper, but they are the first to show that you can take a smaller English focussed language model (LLaMa 7B,13B) and make it into a translation model with equivalent performance to a large LLM (GPT 3.5 &4) using relatively small amounts of monolingual (20B?) and parallel data (1B) in a fine-tuning step. The paper is generally clearly written but some important details are missing, some claims are not entirely supported, and there are some typos.

In more detail:
They first show that large models (GPT3.5) do extremely well translating into and out of  5 diverse languages and English, mostly high resource except for Icelandic. They then select the best performing smaller model (7B parameters) and experiment on fine-tuning to the translation task using instructions. They show that LLama27B quickly maxes out after 10k examples, whereas MPT-7B continues improving until the final datapoint at 20M.
They then experiment with using monolingual data in pretraining, and finetuning with instructions, and call the resulting model ALMA. They show that with a relatively small multilingual (but not parallel) pretraining dataset 1B, they get significant gains in translation quality. Also they used high quality parallel data for finetuning and experimented with full fine-tuning and LORA fine-tuning.

**Strengths:**

They show large improvements in the translation capabilities of the most useful size of models (7B,13B) with very affordable limited fine-tuning and data.
This is a useful paper for people working in machine translation to see what works in fine-tuning large language models for the translation task.

**Weaknesses:**

There is not a lot of novelty in the approach - either in training or modelling. I am not sure that the "New paradigm" title is justified.
I have not learned much from reading the paper - it is still not clear what the contribution of the monolingual vs parallel training data is. It is also not clear whether the good performance of the trained models is due to the reduced number of non English languages (5) vs other models (NLLB, GPT3.5,4).
I am also not sure these results (improvement over LLaMa7B with fine-tuning) would hold if you used few-shot - and it would have been a very easy experiment to conduct.
The paper writing is not particularly clear (see questions for details).

**Questions:**

They claim/state things which are either not entirely correct or are overstated:
"Both monolingual data fine-tuning and human-written data" - I think they mean parallel data here - both monolingual and translated data are human-written.
This claim is not correct: "We demonstrate that LLMs, such as LLaMA-2-7B, do not voraciously consume parallel data. " What they demonstrate is that for fine-tuning LlaMa-2-7B does not improve much beyond 10k examples. However, their other model MPT-7B does keep improving and does not max out even at the 20M mark.
The citation for this claim in the intro: "they still fall short in translation for low-resource languages" Zhang et al. is wrong. They do not look at low resource languages, only experiments with English, Chinese and German.
They conclude: "From our observations, LLMs should not adopt the same training approach as earlier models—whether randomly initialized or pre-trained—that rely heavily on vast amounts of training data" but do not specify that this is just for the fine-tuning LLaMa - it is a too broad claim to make.

Some things are not explained or described:
They do not explain why they selected MPT for experiments in 3.2, and more importantly they do not discuss why it performs contrary to their claimed results - that LLMs to not voraciously consume parallel data.
Also for Section 5 they do not say how much parallel data is used for fine-tuning and what ratios of parallel data are we using - same as the monolingual data? This really should be detailed in the main paper.
This caption is confusing "Figure 5: The average performance of ALMA-7B at the completion of each 1B-token fine-tuning". Is this without the instruction fine-tuning? How do these numbers compare to Tables 1 and 2? I can't figure this out due to the differences in the data - but it seems like the instruction fine-tuning makes little/no difference here. (Fig5) 85.28 COMET vs. (84.12 + 86.27) / 2 (Table 1 and 2) for the ALMA7B?

There are a number of typos and inconsistencies that need to be polished for final submission:
The ALMA models are called AMLA,
Typo:  "As expected, it tends up with a similar performance ".
The graphics are not very consistent in the paper and don't look clean or very legible. Figure 5 is particularly  hard to read.
Results are not very structured. For some (2.2) use NLLB and other not (3.1) .
Many different result formats vertical/horizontal bar, line, table - it is harder to read and looks messy.

---

> ### Author Response · Authors · 2023-11-17
> **Official Rebuttal 1 by Authors**
>
> We genuinely appreciate the valuable feedback provided by the reviewer and have addressed them in a point-by-point manner below. We are more than willing to engage in further discussions with the reviewers should any follow-up questions arise.
>
> In responding to your concerns, we wish to first address a potential misunderstanding. The ALMA models underwent a two-stage fine-tuning process. Initially, they were fine-tuned using monolingual data, with ALMA-7B being trained on 20 billion tokens and ALMA-13B on 12 billion tokens. Next, a fine-tuning was conducted on a smaller set of human-written parallel data. This parallel dataset comprised **58,000 parallel sentences spanning 10 languages**. We believe there is a misalignment between our description of this process and the reviewer's interpretation. Our intention here is to clarify this point to prevent any further misunderstanding.
>
> **Regarding unclear parts in *Weakness***
> >There is not a lot of novelty in the approach - either in training or modelling. I am not sure that the "New paradigm" title is justified
>
> We are grateful for your feedback and would like to provide further clarification. “The new paradigm” specifically addresses machine translation in a new method, rather than proposing a general methodology. We acknowledge that the concepts of 'modeling' (using a decoder-only model) and 'training' (through causal language modeling) are not novel in themselves. However, applying these techniques to machine translation is a significant innovation (Note that we discuss three ‘training’ method for translation: casual language modeling, prefix language modeling, and mixture of denoisers in Appendix A, but the simplest CLM performs the best).  Traditionally, machine translation has primarily relied on processing **vast quantities (millions)** of parallel sentences using an **encoder-decoder** architecture. The application of decoder-only models to this field is basically unexplored and has rarely outperformed the conventional encoder-decoder approach.
>
> The remarkable performance of models like ChatGPT has sparked interest in decoder-only large language models (LLMs) for various applications. There must be potential for machine translation. However, most recent study on machine translation for LLMs fails to achieve comparative performance to conventional encoder-decoder models. This area remains largely untapped, posing critical questions:
>
> - How should we train decoder-only models for translation tasks which were traditionally training on encoder-decoder architecture?
> - Given that LLMs are already extensively trained, is there still a need for large volumes of parallel data for additional training? Could this potentially impair the model's effectiveness?
> - Considering that LLMs are predominantly trained in English, how can we enhance their proficiency in other languages to improve translation quality?
>
> Our paper seeks to address these questions, proposing a novel and efficient training methodology for machine translation. **We are the first to utilize 7B/13B LLMs for machine translation, and achieve performance comparable to state-of-the-art encoder-decoder models.** This breakthrough lays the groundwork for future research in integrating LLMs with machine translation, marking an exciting new direction in the field.
>
> > it is still not clear what the contribution of the monolingual vs parallel training data is
>
> Thank you for the opportunity to further elucidate the roles of monolingual and parallel data in our training process. Although these aspects are presented in Table 3 of our ablation study, we recognize the need for additional clarity:
>
> - Contribution of Monolingual Data: As indicated in Table 3, when the same parallel data fine-tuning is applied (or when no parallel data training is used), models fine-tuned with monolingual data demonstrate notable improvements. This is particularly evident in translations from English to other languages (e.g., in Table 3, compare row 4 with row 8, where the COMET score for **en→xx translation jumps from 76.52 to 86.49)**.
>
> - Contribution of Parallel Data: Also detailed in Table 3, models exhibit enhanced performance when trained with higher-quality parallel data,  under the constraint that the monolingual data training remains constant (or is absent). This trend is observable as we move from row 2 to row 4 in the table. For instance, the **COMET score for en→xx translation progressively increases (73.89 in row 2, 74.35 in row 3, and 76.52 in row 4).**
>
> We are grateful for the feedback highlighting this ambiguity. In our revised manuscript, we will ensure a more comprehensive and explicit discussion of these findings.

---

> > ### Author Response · Authors · 2023-11-17
> > **Official Rebuttal 2 by Authors**
> >
> > >It is also not clear whether the good performance of the trained models is due to the reduced number of non English languages (5) vs other models (NLLB, GPT3.5,4)
> >
> > Thank you for your insightful observation! We believe that the reviewer may be referencing the "curse of multilinguality," a phenomenon where training a model with an increasing number of languages can lead to diminished performance in high-resource languages [1]. However, the primary focus of our paper is on developing robust translation models based on decoder-only large language models (LLMs), rather than exploring how the inclusion of multiple languages affects the performance of the ALMA translation model.
> >
> > As noted in our paper, comparisons with models like GPT-4 or NLLB are not meant to be direct or fair benchmarks but rather serve as a context to gauge the performance of our models in relation to these well-known systems. It's important to highlight that the training data for models like GPT3.5/4 and other moderate-sized LLMs, such as LLaMA and MPT, **often lack detailed language labeling**. For instance, LLaMA-2's training data is composed of 90% English content and approximately 8% in unidentified languages (as detailed in Table 10 of their paper [2]). This lack of precise language identification complicates any analysis regarding the impact of the number of languages trained on these models, and makes it challenging to study the curse of multilinguality in this context.
> >
> > While acknowledging that comparisons between ALMA and models like NLLB or GPT3.5/4 are not entirely equitable due to differences in model size and training data, we do recognize the potential interest in investigating how the number of languages trained might affect ALMA's performance. This remains an intriguing topic for future research, albeit outside the scope of our current study.
> >
> > Reference:
> >
> > [1] Conneau A, Khandelwal K, Goyal N, Chaudhary V, Wenzek G, Guzmán F, Grave E, Ott M, Zettlemoyer L, Stoyanov V. Unsupervised cross-lingual representation learning at scale. arXiv preprint arXiv:1911.02116. 2019 Nov 5.
> >
> > [2] Touvron H, Martin L, Stone K, Albert P, Almahairi A, Babaei Y, Bashlykov N, Batra S, Bhargava P, Bhosale S, Bikel D. Llama 2: Open foundation and fine-tuned chat models. arXiv preprint arXiv:2307.09288. 2023 Jul 18.
> >
> > >I am also not sure these results (improvement over LLaMa7B with fine-tuning) would hold if you used few-shot - and it would have been a very easy experiment to conduct.
> >
> > Thank you for raising this intriguing question. We have indeed conducted the experiments you refer to, and we would like to provide further details:
> >
> > - 5-Shot Learning Performance for LLaMA-2-13B: As presented in Table 6 of our paper, the effectiveness of few-shot learning is influenced by the quality of the provided examples (shots). We observed that higher-quality shots generally lead to improved performance. However, there is a notable performance gap when comparing the LLaMA-2-13B with our ALMA-13B-LoRA model. For instance, the average COMET score for English to various languages (en→xx) is 74.82 for LLaMA-2-13B, significantly lower than the 87.00 achieved by the ALMA-13B-LoRA model.
> >
> > - 5-Shot Learning for LLaMA-2-13B After Monolingual Fine-Tuning: After undergoing monolingual fine-tuning with 12 billion tokens, the results for 5-shot learning are detailed in Table 11, where it still does not match the performance of fully supervised learning. For example, in the case of en→xx, the average COMET score for 5-shot learning is 85.24, which is still lower than the 86.37 score achieved through supervised learning.
> >
> > These findings highlight the limitations of few-shot learning in certain contexts. We have also explored the intriguing concept of using prompts written in target languages, as detailed in Table 6 of our paper. We encourage the reviewer to review this section for a comprehensive understanding of our findings of in-context learning.

---

> > > ### Author Response · Authors · 2023-11-17
> > > **Official Rebuttal 3 by Authors**
> > >
> > > **Regarding concerns in Questions:**
> > > >They claim/state things which are either not entirely correct or are overstated: "Both monolingual data fine-tuning and human-written data" - I think they mean parallel data here - both monolingual and translated data are human-written.
> > >
> > > Thank you for your feedback. To provide further clarification, we did not verify whether the monolingual data was human-written; we only confirmed this for the parallel data. This distinction might relate to the initial misunderstanding we addressed earlier.
> > >
> > > >This claim is not correct: "We demonstrate that LLMs, such as LLaMA-2-7B, do not voraciously consume parallel data. " What they demonstrate is that for fine-tuning LlaMa-2-7B does not improve much beyond 10k examples. However, their other model MPT-7B does keep improving and does not max out even at the 20M mark
> > >
> > > Thank you for your meticulous review of our claim. We would like to delve deeper into this discussion. It's true that the MPT-7B model shows continued improvement with the addition of more parallel data. Our hypothesis is that this improvement may be due to MPT-7B not being as proficient in English and Russian as LLaMA-2-7B. To illustrate this, consider an extreme scenario where a large language model (LLM) starts with a random initialization; in such a case, its performance would undoubtedly improve with the inclusion of more parallel datasets. In our revised manuscript, we will make it clear that not all LLMs necessarily require additional parallel data for improvement, particularly those that are not as well-trained on target languages initially.
> > >
> > > >The citation for this claim in the intro: "they still fall short in translation for low-resource languages" Zhang et al. is wrong
> > >
> > > Thank you for your thorough review of our citations. We concur with your observation that the reference to Zhang et al. is not appropriate in this context. We will ensure its removal in the forthcoming version of our manuscript.
> > >
> > > >They do not look at low resource languages, only experiments with English, Chinese and German.
> > >
> > > Thank you for your attention to this matter. To clarify, our paper encompasses studies on five languages in addition to English. Among these, four are high-resource languages (German, Czech, Chinese, and Russian), and one is a **low-resource language (Icelandic)**. If we have misunderstood the focus of your concern, please provide us with additional guidance for a more accurate understanding.
> > >
> > > >They conclude: "From our observations, …, ,rely heavily on vast amounts of training data" but do not specify that this is just for the fine-tuning LLaMa - it is a too broad claim to make.
> > >
> > > Thank you for your valuable feedback. We are dedicated to ensuring that our revised manuscript accurately reflects our findings, avoiding any unintended exaggeration. To clarify, our refined claim will state: "From our observations, **LLaMA-2 (potentially other well-trained LLMs)** should not should not adopt the same training approach as earlier models—whether randomly initialized or pre-trained—that rely heavily on vast amounts of training data."
> > >
> > > > They do not explain why they selected MPT for experiments in 3.2, and more importantly they do not discuss why it performs contrary to their claimed results - that LLMs to not voraciously consume parallel data.
> > >
> > > Thank you for the opportunity to provide more clarity. We chose MPT and LLaMA-2 for our experiments in Section 3.2 specifically because they demonstrated the best performance in zero-shot translation among six popular large language models (LLMs), as noted in the concluding paragraph of Section 2. We acknowledge your feedback and will ensure a more detailed explanation of this choice in our revised manuscript.
> > >
> > > >Also for Section 5 they do not say how much parallel data is used for fine-tuning and what ratios of parallel data are we using - same as the monolingual data?
> > >
> > > Thank you for allowing us to clarify further. In fact, we have specified the size of the parallel dataset in the first sentence of Section 5.1: “For our parallel training data, we collect human-written test datasets from WMT’17 to WMT’20, plus the development and test sets from Flores-200, resulting in a total of 58K training examples across all languages.” Furthermore, for a more comprehensive understanding, we have included detailed statistical information about the parallel data in Table 5 of our paper.

---

> > > > ### Author Response · Authors · 2023-11-17
> > > > **Official Rebuttal 4 by Authors**
> > > >
> > > > >This caption is confusing "Figure 5: The average performance of ALMA-7B at the completion of each 1B-token fine-tuning". Is this without the instruction fine-tuning? How do these numbers compare to Tables 1 and 2?
> > > >
> > > > Thank you for your questions, and we're glad to provide further clarification. As mentioned at the start of our rebuttal, the ALMA models undergo a two-stage fine-tuning process. The first stage involves fine-tuning with monolingual data, followed by the second stage, which uses human-written parallel data for fine-tuning. We conduct an evaluation of the ALMA model after each phase of fine-tuning with every 1 billion (1B) monolingual tokens. For instance, when we refer to “3B” in Figure 5, it indicates that the ALMA model has been fine-tuned on 3 billion monolingual tokens, followed by fine-tuning with human-written parallel data. Results in Table 1 and 2 are models fine-tuned by 20 billion tokens followed by parallel data fine-tuning.
> > > > Regarding the numbers in comparison to those in Table 1 and 2, the reviewer can refer to the row labeled “ALMA-7B”. There, the reviewer will find consistent results: (86.49 + 84.08) / 2 = 85.28.

---

### Official Review · Reviewer_vUYo · 2023-11-01

**Soundness:** 3 good
**Presentation:** 4 excellent
**Contribution:** 3 good
**Rating:** 6
**Confidence:** 4

**Summary:**

This paper focuses on improving the translation capability of large language models (LLMs). The authors find that LLMs do not require a large amount of parallel data as traditional models do to achieve decent translation quality. Accordingly, they propose a new training recipe including two stages: firstly finetune LLM on monolingual data and then finetune the model on a small set of high-quality parallel data. Experiments with LLAMA-2 show promising performance even on par with GPT-3.5.

**Strengths:**

1) Propose a simple training recipe for LLMs for translation tasks: finetuning first on monolingual data and then on small high-quality parallel data.
2) Demonstrate impressive performance across 5 language pairs with LLAMA-2.

**Weaknesses:**

1) The statement of "paradigm shift" is somehow overestimated.
2) The few-shot prompting results are highly undervalued.
3) The proposed recipe might not apply to other LLMs and languages.

**Questions:**

Firstly, the authors claim the finding/proposal is a "paradigm shift" as highlighted in the title, which might be inadequate. The recipe generally follows the pretraining-finetuning paradigm which has already been well-established since BERT and mBart/T5. In addition, similar solutions have already been used in prior studies, such as BigTranslate.

Secondly, the few-shot results are largely undervalued. As shown in Table 11, the quality gap between the HW 5-shot and the proposal is only 0.2 COMET on XX->En, although the finetuning used a large amount of monolingual corpus and more parallel data. Few-shot performance should be able to be further enhanced via beam search and optimized prompt construction, and it should be used as the fair baseline rather than 0-shot prompting. It's also misleading to state in Appendix I that "ICL substantially underperforms our stage-2 finetuning".

Lastly, finetuning performance is highly dependent on pretraining conditions and the downstream task. Intuitively, when the downstream tasks highly correlate with the pretraining, the demand for a large volume of the supervised corpus is reduced as highlighted in this paper. However, as the MPT-7B performance indicates in Figure 4, when the correlation is low, adding more supervised data is almost always helpful. In other words, the findings in this paper might not be generalization. Do you also have the finetuning results for MPT-7B? Does the recipe also apply to low-resource languages like Gu?

---

> ### Author Response · Authors · 2023-11-17
> **Official Rebuttal 1 by Authors**
>
> We genuinely appreciate the valuable feedback provided by the reviewer and have addressed them in a point-by-point manner below. We are more than willing to engage in further discussions with the reviewers should any follow-up questions arise.
>
> **Regarding your concern about statement:**
> >The statement of "paradigm shift" is somehow overestimated.
>
> We are grateful for your feedback and would like to provide further clarification. “The new paradigm” specifically addresses machine translation in a new method, rather than proposing a general methodology. While the general concept of "pre-training + fine-tuning" is indeed widely used in NLP tasks, it represents a broad strategy rather than a specific methodology. We acknowledge that the concepts of 'modeling' (using a decoder-only model) and 'training' (through causal language modeling) are not novel in themselves. However, applying these techniques to machine translation is a significant innovation (Note that we discuss three ‘training’ method for translation: casual language modeling, prefix language modeling, and mixture of denoisers in Appendix A, but the simplest CLM performs the best).  Traditionally, machine translation has primarily relied on processing vast quantities (millions) of parallel sentences using an encoder-decoder architecture. The application of decoder-only models to this field is basically unexplored and has rarely outperformed the conventional encoder-decoder approach.
>
> The remarkable performance of models like ChatGPT has sparked interest in decoder-only large language models (LLMs) for various applications. There must be potential for machine translation. However, most recent study on machine translation for LLMs fails to achieve comparative performance to conventional encoder-decoder models. This area remains largely untapped, posing critical questions:
>
> - How should we train decoder-only models for translation tasks which were traditionally training on encoder-decoder architecture?
> - Given that LLMs are already extensively trained, is there still a need for large volumes of parallel data for additional training? Could this potentially impair the model's effectiveness?
> - Considering that LLMs are predominantly trained in English, how can we enhance their proficiency in other languages to improve translation quality?
>
> Our paper seeks to address these questions, proposing a novel and efficient training methodology for machine translation. We are the first to utilize 7B/13B LLMs for machine translation, and achieve performance comparable to state-of-the-art encoder-decoder models. This breakthrough lays the groundwork for future research in integrating LLMs with machine translation, marking an exciting new direction in the field.
>
> We recognize that previous studies, such as BigTranslate, have implemented similar fine-tuning approaches. However, there are significant differences between their methods and ours, particularly in acknowledging the negative impacts of excessive parallel data and the importance of data quality. Their approach resulted in performances that did not meet our expectations. Our goal is to provide an accurate and effective training "recipe" for decoder-only LLMs in machine translation, ensuring high performance. In our revised manuscript, we commit to clearly delineating our method's relationship to the broader "pre-training and fine-tuning" approach, emphasizing the unique aspects of our strategy and avoiding any unintentional overstatements.

---

> > ### Author Response · Authors · 2023-11-17
> > **Official Rebuttal 2 by Authors**
> >
> > **Regarding your concern about Few-shot learning:**
> > >The few-shot prompting results are highly undervalued.
> >
> > Thank you for your valuable feedback and for taking the time to review the appendix. We would like to clarify an important aspect: the model evaluated for few-shot learning is not the standard LLaMA-2-13B, but rather LLaMA-2-13B after fine-tuning with 12 billion monolingual data tokens. This means that both our supervised fine-tuning and few-shot learning methods are built upon a model that has extensively utilized monolingual corpus. For contexts where monolingual data fine-tuning is not applied, we kindly direct you to Table 6 (Appendix E). In this table, it is evident that a 5-shot fine-tuning approach applied solely on LLaMA-2-13B falls behind the ALMA-13B-LoRA model by 12.18 COMET points on en$\rightarrow$xx direction.
> >
> > However, we agree with the reviewer's suggestion that we need to exercise caution with our wording, particularly since few-shot learning performance can be significantly influenced by the quality of the prompts used. Nonetheless, supervised fine-tuning still offers certain advantages under these conditions:
> > - Performance Edge: Supervised fine-tuning, albeit marginally, continues to outperform few-shot learning, especially in English-to-other language (en$\rightarrow$xx) translations. This observation is consistent with findings from several previous studies ([1], [2]).
> > - Stability of Results: The performance of few-shot learning tends to vary considerably, heavily dependent on the quality of examples (shots) included in the prompt, as demonstrated in Table 11.
> > - Efficiency in Resource Use: Few-shot learning typically requires a longer prefix, consuming more memory. If we consider a hypothetical scenario where each input token incurs a cost, supervised fine-tuning would be more economical in terms of resource utilization.
> >
> >
> > Reference:
> >
> > [1] Liu H, Tam D, Muqeeth M, Mohta J, Huang T, Bansal M, Raffel CA. Few-shot parameter-efficient fine-tuning is better and cheaper than in-context learning. Advances in Neural Information Processing Systems. 2022 Dec 6;35:1950-65.
> >
> > [2] Mosbach M, Pimentel T, Ravfogel S, Klakow D, Elazar Y. Few-shot Fine-tuning vs. In-context Learning: A Fair Comparison and Evaluation. arXiv preprint arXiv:2305.16938. 2023 May 26.]

---

> > > ### Author Response · Authors · 2023-11-17
> > > **Official Rebuttal 3 by Authors**
> > >
> > > **Regarding your concern about generalization:**
> > > >The proposed recipe might not apply to other LLMs and languages.
> > >
> > > > However, as the MPT-7B performance indicates in Figure 4, when the correlation is low, adding more supervised data is almost always helpful.
> > >
> > > Thank you for your insightful comment. We concur with your observation that the effectiveness of downstream tasks is closely linked to the pre-training stage. This understanding actually underscores the importance of non-English monolingual fine-tuning for these models. LLMs mainly trained in English are likely to exhibit performance trends similar to the MPT model, especially in languages like Russian, which they have not been exposed to during training. In such cases, any improvements in these models are solely reliant on the Russian knowledge extracted from parallel data. However, learning from parallel data can wash out the knowledge learned during the pre-training phase — to the extent that a model initialized randomly could achieve comparable performance to MPT after being trained with 20 million parallel data sentences. The primary aim of monolingual fine-tuning is to equip the model with a general understanding of a new language. Following this, parallel data fine-tuning is employed to guide the model in accurately utilizing this knowledge for effective translation. Our suggested approach is designed to be **universally applicable across all LLMs and languages**. It involves fine-tuning the model on monolingual data of the target language, regardless of its resource availability, and subsequently fine-tuning on the corresponding parallel data. This method ensures that the model not only learns the new language but also becomes proficient in applying this knowledge for translation tasks.
> > >
> > > >In other words, the findings in this paper might not be generalization. Do you also have the finetuning results for MPT-7B? Does the recipe also apply to low-resource languages like Gu?
> > >
> > > To demonstrate the generalization of our methods to other large language models (LLMs), we extended our experiments to Mistral-7B, another well-known but recently released model. We fine-tuned Mistral-7B using 18 billion tokens of monolingual data, which included a mix of five new languages as detailed in our paper. We then evaluated its performance on the same multilingual test set used for our previous experiments. The results, as presented below, show that ALMA models, whether based on LLaMA-2-7B or Mistral-7B, are capable of achieving comparable levels of performance. This indicates the potential of our fine-tuning approach to be effectively generalized across different LLM architectures.
> > >
> > > |                | Avg. en>xx |       | Avg. xx>en |       |
> > > |----------------|------------|-------|------------|-------|
> > > |                | BLEU       | Comet22 | BLEU       | Comet22 |
> > > | ALMA-LLaMA-7B-LoRA | 29.78 | 86.37 | 34.31 | 84.12 |
> > > | ALMA-Mistral-7B-LoRA | 30.78 | 86.48 | 34.07 | 84.09 |
> > >
> > > To demonstrate the adaptability of our methods to low-resource languages, specifically Gujarati (gu), we initially fine-tuned LLaMA-2-7B using 0.1 billion Gujarati tokens and subsequently fine-tuned it on the Flores-200 development data. The model was then evaluated using the Flores-200 devtest dataset. The results of this experiment are detailed below. We compared the performance of our fine-tuned model against the zero-shot performance of LLaMA-2-7B and the outputs from BigTranslate. These comparisons revealed that while the other models produced some nonsensical outputs, our fine-tuned model significantly outperformed them, showcasing the effectiveness of our method. It's important to note that this was a preliminary experiment, constrained by time limitations, and does not represent the peak potential performance for Gujarati. We anticipate that the performance could be significantly enhanced with access to more monolingual data, higher-quality data, and more refined training practices. This experiment serves as evidence that our method can be successfully extended to low-resource languages like Gujarati.
> > >
> > > |                | en>gu       |       | gu>en       |       |
> > > |----------------|-------------|-------|-------------|-------|
> > > |                | BLEU        | COMET22 | BLEU        | COMET22 |
> > > | LLAMA-2-7B, zero-shot | 0.48      | 38.88 | 0.18      | 39.33  |
> > > | BigTranslate   | 0.23      | 43.80 | 1.87      | 51.09  |
> > > | ALMA-7B-GU     | **11.31**     | **79.42** | **15.05**     | **66.60**  |

---

### Official Review · Reviewer_TXCQ · 2023-11-01

**Soundness:** 4 excellent
**Presentation:** 4 excellent
**Contribution:** 4 excellent
**Rating:** 8
**Confidence:** 4

**Summary:**

The paper is motivated by that the translation performance of LLMs are not as good as other tasks compared to the task-specific methods. To improve the translation capabilities of the moderate LLMs, it proposes a fine-tuning paradigm which is firstly fine-tuning on monolingual data followed by subsequent fine-tuning on a small set of high-quality parallel data. It turns out a huge gain in translation quality compared to the zero-short performance across 10 translation directions from the WMT21 and WMT22.

**Strengths:**

The paper is clearly written and provides many insights. Using LLM to boost the translation quality is an interesting and important topic. It proposes a novel fine-tuning paradigm to let the moderate size LLMs better at translation. Many analyses should be very helpful to the NLP and ML community.

**Weaknesses:**

The paper is mainly focusing on improve the translation quality of LLMs. It'd be better to compare more with the encoder-decoder translation models and shed light on the best practice of translation itself.

**Questions:**

How's your fine-tuned models compared with the dedicated encoder-decoder based translation models in similar size? Please discuss in both high and low resource settings.

If fine-tuning with large amount of parallel data is not optimal, how about using the during pre-training phase? If targeting at the highest translation performance, what's your take? Please include the dedicated translation model in the discussion.

What would be the results of evaluating on some out-of-domain test sets?

---

> ### Author Response · Authors · 2023-11-17
> **Official Rebuttal 1 by Authors**
>
> We genuinely appreciate the valuable feedback provided by the reviewer and have addressed them in a point-by-point manner below. We are more than willing to engage in further discussions with the reviewers should any follow-up questions arise.
>
> **Regarding your question about dedicated encoder-decoder model**:
> >How's your fine-tuned models compared with the dedicated encoder-decoder based translation models in similar size?
>
> Thank you for your insightful question! Most online translation models are generally smaller, often adhering to the transformer-big architecture [1], which results in around 300 million parameters. We'd like to clarify that in our paper, the NLLB-200 model serves as a benchmark for one of the strongest encoder-decoder based translation models available that is comparable in size to our ALMA models. The NLLB-54B's base model has 3 billion parameters, with additional parameters derived from a Mixture of Experts (MoE) approach. Our research demonstrates that for high-resource languages such as Chinese, German, Russian, Czech, and the low-resource language Icelandic, our ALMA models either match or outperform the dedicated encoder-decoder model NLLB-54B. However, we acknowledge that for very low-resource languages, where even monolingual data is limited, further research is required to optimize positive cross-lingual transfer. We plan to explore these avenues in future studies.
>
> Reference:
> [1] Vaswani A, Shazeer N, Parmar N, Uszkoreit J, Jones L, Gomez AN, Kaiser Ł, Polosukhin I. Attention is all you need. Advances in neural information processing systems. 2017;30.
>
> **Regarding your question about using parallel data for pre-training:**
> >If fine-tuning with large amount of parallel data is not optimal, how about using them during the pre-training phase?
>
> Thank you for your intriguing question. In fact, at an early stage in our research, we experimented with treating parallel data as monolingual data during the pre-training phase. Specifically, we used 20 million English-Russian parallel sentences, amounting to a total of 1.8 billion tokens. Our initial approach involved fine-tuning the LLaMA-2-7B model on these 1.8 billion tokens, mimicking the pre-training process, followed by further fine-tuning on English to Russian translation using the Flores development and test datasets (comprising approximately 2,000 sentences in total). To ensure a fair comparison, we also conducted an experiment where we fine-tuned LLaMA-2-7B on 1.8 billion tokens from the Oscar dataset, consisting of both English and Russian data, and this was also followed by fine-tuning on the same English-Russian parallel data. The outcomes of these comparative tests are presented below:
>
> |                              | BLEU  | Comet22 |
> |------------------------------|-------|---------|
> | Parallel data as monolingual data | 25.26 |  85.29  |
> | Oscar as monolingual data    | **26.43** |  **86.36**  |
>
> The results indicate that treating parallel data as monolingual data is beneficial, but it does not yield as favorable outcomes as using authentic monolingual data, such as that from the Oscar dataset. We hypothesize that the advantage of using Oscar's monolingual data lies in its document-level context, which enables ALMA models to learn more natural expressions in Russian. In contrast, treating parallel data as monolingual data limits learning to sentence-level information, which may not provide the same depth of linguistic understanding and fluency.

---

> > ### Author Response · Authors · 2023-11-17
> > **Official Rebuttal 2 by Authors**
> >
> > This phenomenon potentially offers insights into your next question:
> > >If targeting at the highest translation performance, what's your take?
> >
> > A short answer is, the key to effective LLM-based translation lies in acquiring general language knowledge and establishing good bilingual alignment. A proficient translation model should first thoroughly understand the target language. However, many LLMs, primarily trained in English, often fall short in comprehending other languages. This deficiency is a primary factor behind the subpar translation performance of moderate-sized LLMs. Once a robust grasp of the target language is achieved, the next critical step is to ensure proper (bilingual) alignment. This alignment steers the model towards generating the desired translation output.  Our supervised fine-tuning with high-quality data is a viable approach to achieve this alignment, but there could potentially be more effective methods to further enhance this alignment process.
> >
> > To illustrate the differences in translation quality between a dedicated encoder-decoder model (NLLB-54B, trained on sentence-level parallel data) and our ALMA-13B-LoRA model (which learns more general language information + bilingual alignment), let's consider an example involving specific Chinese internet slang. The terms “八卦” (bā guà) and “实锤” (shí chuí) literally translate to “eight trigrams" and "solid hammer" respectively, but in slang, they mean “gossip" and "solid truth." When translating the phrase “这个八卦实锤了”, most translation systems, including NLLB-54B, face challenges. However, our ALMA-13B-LoRA model exhibits a more natural and accurate understanding in its translation:
> >
> > - Source Chinese sentence: "这个八卦实锤了"
> > - NLLB-54B translation: "This is a real shame."
> > - ALMA-13B-LoRA translation: "The gossip is true."
> >
> > **Regarding your question about out-of-domain testing:**
> > >What would be the results of evaluating on some out-of-domain test sets?
> >
> > Thank you for providing an insightful question! We actually have conducted such an experiment at Appendix E and we encourage the reviewer to take a look with any chances. Here, we only compare the ALMA model fine-tuning only Flores to the one fine-tuned by Flores+WMT data. Since the test data is WMT’22 (even though it has 4 difference domains), we think WMT training data we use is “in-domain” data, and Flores is out-of-domain data. Our results show that training with out-of-domain data may lead the lexical match drop (BLEU score down), while COMET scores are similar or even better.

---

### Author Response · Authors · 2023-11-21

Dear Reviewers,

We hope this message finds you well. As we approach the end of the discussion period, we want to reach out regarding our recent rebuttal. Should you find any aspects of our rebuttal compelling or appropriate, we would be grateful if you could consider reflecting this in your updated score. We recognize and appreciate the extensive responsibilities you manage, and your time and effort in reviewing our work is highly valued. Your insights and critiques are not only helpful but crucial to our progress.

Thank you once again for your time and consideration.

---

### Meta-Review · Area_Chair_r3bA · 2023-12-15

**Metareview:**

This paper focuses on improving the translation capability of large language models (LLMs) via fine-tuning. The authors conduct experiments with a two stages fine-tuning approach: firstly they fine-tune a LLM on monolingual data and then fine-tune the model on a small set of high-quality parallel data. Experiments with LLAMA-2 show large performance improvements even on par with much larger LLMs like GPT-3.5. However, the approach is not new, the strength of the paper are the experimental results that demonstrate that fine-tuning is much more effective than what people did before with BERT/BART/T5 or BigTranslate. The "New paradigm" title is not justified and overestimated the contribution of the paper. Overall, the paper contributes to the community with experimental results that are interesting, but do not present new methods. We highly recommend adapting the paper to the actual contributions of the paper. There are also a couple of papers doing fine-tuning of LLMs that are concurrent work and have been published during the review period.

Citations: [1] MBR and QE Finetuning: Training-time Distillation of the Best and Most Expensive Decoding Methods

**Justification For Why Not Higher Score:**

The paper oversestimated their impact and contributions. I think it is a good paper, but they have to be honest about their contributions.

**Justification For Why Not Lower Score:**

The paper shows interesting experiments and there are no major flaws in their experiments.

---

### Decision · Program_Chairs · 2024-01-16

Accept (poster)